# Social inequalities, length of hospital stay for chronic conditions and the mediating role of comorbidity and discharge destination: A multilevel analysis of hospital administrative data linked to the population census in Switzerland

**Lucy Bayer-Oglesby** *, **Andrea Zumbrunn, Nicole Bachmann, on behalf of the SIHOS Team¶**

Institute for Social Work and Health, School of Social Work, University of Applied Sciences and Arts Northwestern Switzerland, Olten, Switzerland

¶ Membership of the SIHOS Team is provided in the Acknowledgments.
* lucy.bayer@fhnw.ch

## Abstract

Social factors are recognized determinants of morbidity and mortality and also have an impact on use of medical services. The objective of this study was to assess the associations of educational attainment, social and financial resources, and migration factors with length of hospital stays for chronic conditions. In addition, the study investigated the role of comorbidity and discharge destination in mediating these associations. The study made use of nationwide inpatient data that was linked with Swiss census data. The study sample included n = 141,307 records of n = 92,623 inpatients aged 25 to 84 years, hospitalized between 2010 and 2016 for a chronic condition. Cross-classified multilevel models and mediation analysis were performed. Patients with upper secondary and compulsory education stayed longer in hospital compared to those with tertiary education (β 0.24 days, 95% CI 0.14–0.33; β 0.37, 95% CI 0.27–0.47, respectively) when taking into account demographic factors, main diagnosis and clustering on patient and hospital level. However, these effects were almost fully mediated by burden of comorbidity. The effect of living alone on length of stay (β 0.60 days, 95% CI 0.50–0.70) was partially mediated by both burden of comorbidities (33%) and discharge destination (30.4%). (Semi-) private insurance was associated with prolonged stays, but an inverse effect was observed for colon and breast cancer. Allophone patients had also prolonged hospital stays (β 0.34, 95% CI 0.13–0.55). Hospital stays could be a window of opportunity to discern patients who need additional time and support to better cope with everyday life after discharge, reducing the risks of future hospital stays. However, inpatient care in Switzerland seems to take into account rather obvious individual needs due to lack of immediate support at home, but not necessarily more hidden needs of patients with low health literacy and less resources to assert their interests within the health system.

**Data Availability Statement:** The data underlying the results presented in the study are available form the Swiss Federal Statistical Office (SFSO) Sektion Gesundheitsversorgung, Espace de l'Europe10, CH-2010 Neuchâtel, Switzerland, Phone: +41 58 463 67 00, Email: gesundheit@bfs.admin.ch. The supporting information file (S1 File) contains the description of the data sources and the specification of the variables that need to be requested from the SFSO.

**Funding:** This work was supported by the SNSF National Research Programme "Smarter Health Care"(NRP74), project number 4, grant number 407440_167506, applicant LBO. Project and funding description are available at http://www.nfp74.ch/en/projects/in-patient-care/project-bayer-oglesby. The funder had no role in the study design, data collection and analysis, decision to publish, or preparation of the manuscript. The views reported here are the authors' views and do not necessarily reflect the funding organization.

**Competing interests:** The authors have declared that no competing interests exist.

# Introduction

Social factors are major determinants of morbidity and mortality in Europe and worldwide [1, 2]. There is also broad evidence of social differences in the use of inpatient medical services. Those with a low education level [3–5], low health literacy [6], a low income [5, 7] or limited social support [3, 4, 8, 9] are at higher risk for hospitalisation due to chronic conditions. Socially disadvantaged persons show elevated hospitalisation risks particularly for ambulatory care sensitive conditions (ACSC) such as diabetes, congestive heart failure (CHF), chronic obstructive pulmonary disease (COPD) and asthma [3, 5, 7]. Migration status is also associated with differential utilization of health services: Migrants of the first and the second generation tend to consult general practitioners more often and specialists less often than persons without a migration background in Switzerland [10], while migration background has been associated with lower hospitalisation risks [5]. In the migration population of Switzerland, the lack of local language skills is associated with poorer health and more limitations due to health problems [11].

In Switzerland and other countries in central Europe, an increase in the prevalence of *multimorbidity* (coexistence of two or more chronic disorders) in the population aged 50 and over has been observed over the last decade [12]. Several studies describe a social gradient for multimorbidity and comorbidity regarding area level deprivation [13–16], educational attainment [4, 8] and income [17] while social resources are established predictors of morbidity and mortality [18, 19]. In addition, Barnett et al. [13] reported that mental health comorbidity increased with the number of physical disorders and that the onset of multimorbidity occurred 10–15 years earlier in people living in the most deprived areas compared to those in the most affluent areas.

With regard to hospitalisations multimorbidity and comorbidity have been found to be associated with unplanned, preventable and more frequent hospital admissions, particularly in those with COPD, Diabetes and CHF [15, 20–22]. Higher comorbidity scores have been associated with increased number of hospital bed days [23, 24] and with a longer length of hospital stays [25–27]. In another study, the length of hospital stay increased with the number of diagnoses after adjusting for demography and SES [28].

There is evidence from numerous studies that also *lower socioeconomic status* is related to longer hospital stays. While area level socioeconomic disadvantage has been associated with more cumulative bed days [16, 23, 29], Ghosh et al. [30] reported that wealthier patients (according to median income by zip-code) stayed in hospital for a shorter time compared to poorer patients, but that the difference was more pronounced for discharge to go home than for non-home destinations. Studies measuring social factors on the individual level found that low educational attainment was associated with increased numbers of bed-days [4] and that low health literacy [31] and fewer financial resources [28] were associated with a longer length of hospital stay. However, in studies that were able to adjust for demography as well as for indicators of health status such as comorbidity, severity or main diagnosis, the effects of both area level social deprivation [32] and individual education level and income [26] were not significantly associated with the length of stay.

Comorbidity may therefore act as a *mediator* in the association between education level and length of hospital stays. The literature mentioned above suggests that educational attainment is a significant predictor of both comorbidity and length of hospital stay, while comorbidity is a significant predictor of length of hospital stay. Since educational attainment generally precedes the onset of (multiple) chronic conditions and since it is also plausible that a higher burden of comorbidities causes longer hospital stays and not vice versa, a causal pathway can be postulated from education level over comorbidity to length of hospital stay.

Several studies suggest that patients with a *lack of social support* may have to stay in hospital until they are sufficiently independent to cope at home or until a place in another inpatient setting such as rehabilitation, skilled nursing facility care or long-term care has been organized and is available. In a representative sample of the non-institutionalised population of Switzerland, the availability of informal care within the household has been found to significantly reduce the length of hospital stays, independent of whether the support came from a spouse or from other adults [33]. The lack of help at home from the patient's partner has also been found to increase the likelihood of discharge to post-acute care instead of home discharge [34] while transfer to another hospital was associated with longer length of stay (unadjusted) [28]. Particularly among older patients, those living alone have been found to stay longer in hospital [35], to have higher odds of non-home discharge [35] and higher odds of discharge to skilled nursing facility care [36] compared to those living with others.

The literature overview suggests that both, comorbidity and discharge destination may act as mediators in the association between living alone and length of stay. First, social resources are established predictors of morbidity [18] and of prolonged hospital stays [33, 35]. A causal pathway can be postulated from living alone over comorbidity to length of stay, though this pathway may be more complex. Living alone has been associated with better functional status, particularly in the elderly population [35, 37] while chronically ill persons may have difficulty maintaining social contacts and may therefore be living alone [19]. Second, the literature suggests that living alone is a significant predictor of both non-home discharge (e.g., to rehabilitation before returning home alone) and prolonged hospital stay (e.g., until the patient is sufficiently independent to cope at home), while non-home discharge is a significant predictor of prolonged hospital stays (e.g., time needed to organize a transfer and waiting time for the appropriate institution). Since the decision-making process takes place in hospital and precedes the actual time of discharge a plausible causal pathway goes from the type of household over the discharge destination to length of hospital stay and not the other way round.

*Language and cultural barriers* are likely to hamper communication between hospitalized patients and health professionals and may have an impact on the use of diagnostic procedures and treatments and consequently on the length of stay [38, 39]. Language barriers have been found to be related to patient safety risks in hospital care [40], to poorer understanding of discharge instructions [41] and higher risks for readmissions, particularly in patients with heart failure and COPD [42]. Studies on the provision of professional language interpretation in acute care hospitals and length of stay report controversial results: While Lindholm et al. [43] found evidence of shorter stays for use of interpreter service in patients with limited language skills, Abbato et al. [44] observed longer stays for patients provided with interpreter service admitted to the hospital ward but shorter stays for those admitted to ED. In another hospital-based study, patients provided with interpreter service also had longer stays [45]. As possible explanation the authors discuss the selective use of interpreters for medically more complex patients, a phenomenon that has also been observed in Switzerland [46].

This paper addresses the impact of social factors on the length of hospital stay for leading chronic conditions in a high-income country. Unlike previous studies on cumulative bed-days [4, 16, 23, 24, 29], the present analysis of hospital admissions takes into account factors that are related to the hospital stay and that may act as confounders or mediators. These include main diagnosis, comorbidity, treatments, and discharge destination. The analysis makes use of the database of the study "Social Inequalities and Hospitalisations in Switzerland SIHOS", which is part of the Swiss National Research Programme "Smarter Health Care" (NRP74). The SIHOS study investigated social disparities that may manifest at different stages of a hospitalization: before hospital admission [3], during the hospital stay (this paper), at discharge [47] or

after the hospital stay [48]. The focus was on non-communicable chronic diseases (NCD), which accounted for 80 percent of total healthcare costs in Switzerland in 2011 [49].

The SIHOS database combined national hospital administrative data with national population census data in anonymised form on the individual level for the first time in Switzerland. Based upon the linkage of these data sources, the SIHOS database includes information on the social situation, health status and hospital stays of a representative sample of the Swiss population. With the resulting retrospective inpatient cohort, the paper addresses the following questions:

1. Are the social characteristics of inpatients (education level, financial and social resources and factors related to migration) associated with the length of hospital stays for chronic conditions, if demographic factors, health status (main diagnosis and comorbidity), treatment-related factors, discharge destination, variation at hospital level and multiple stays are simultaneously taken into account?

2. Does comorbidity of inpatients and discharge destination act as mediators of the associations between social factors and length of stay, with indirect effects along the pathways i) educational attainment-comorbidity-length of stay, ii) living alone-comorbidity -length of stay and iii) living alone -discharge destination- length of stay?

3. Which of the investigated determinants i) demographic and social factors of inpatients, ii) health status of inpatients iii) treatment-related factors and iv) discharge-destination are the main drivers of length of hospital stays?

Switzerland has a universal health insurance system that is compulsory and that covers ambulatory, outpatient, and inpatient care. As in other countries, the length of hospital stay has gradually decreased in Switzerland over the last decade [50, 51]. Further, the hospital reimbursement system changed in 2012 from a fee-for-service per diem system to a fixed rate per diagnosis-related group system (SwissDRG) [50, 51]. This change went hand in hand with an increase of transfers from acute care to rehabilitation and other institutional care [51]. We assume that these developments may have increased the pressure for premature discharge, particularly of socially disadvantaged patients needing a prolonged stay for a good outcome, but with fewer resources to assert their interests within the health system. In hospitals committed to equity, such as Swiss Hospitals for Equity, though, chronically ill patients with poor health literacy and self-management skills may be provided with additional support that may result in extra hospital days.

## Methods

We follow the Strengthening the Reporting of Observational Studies in Epidemiology (STROBE) guidelines [52] and the The REporting of studies Conducted using Observational Routinely-collected health Data (RECORD) [53] statement as well as Barker's recommendations for reporting cross-classified multilevel models (CCMM) [54].

### SIHOS database: Data sources and study population

The study population of the SIHOS database is defined by the Structural Survey 2010–2014 (Swiss census data, SE). The SE annually provides (reference day 31 December) information on socioeconomic status, migration status, working status and type of household of a representative sample of about 200,000 persons aged 15 years and older and living in private households in all regions of Switzerland. The response rate of the SE is about 87 percent, and the sample corresponds to about 3.5 percent of the Swiss population aged 15 and over. The

sampling procedures of the SE have been described in detail elsewhere [55]. The Hospital Medical Statistics (MS) and the Statistics on Medico-Social Institutions (SOMED) are comprehensive surveys on the use of inpatient care by the Swiss population and should therefore contain all admissions to inpatient institutions of the participants of the Structural Survey.

For the MS/SOMED an anonymous linking code is routinely generated in order to link subsequent admissions of the same person [56, 57]. It is generated through established processes of unidirectional hashing followed by reverse encrypting. Details are described elsewhere [56]. In the framework of the SIHOS study, this anonymous linking code has been generated for the first time for the SE. This allowed us to match on the individual level 1.2 million records from the SE 2010–2014 with 9.6 million records from the MS 2010–2016, 1.0 million records from the SOMED 2010–2016, 0.4 million mortality records from the Swiss Vital Statistics (BEVNAT 2011–2016) and 1.0 million house-moving records from the Population and Household Statistics (STATPOP movements 2011–2016) (Fig 1). For each year of the SE (2010, 2011, 2012, 2013 and 2014) record linkage was performed with all MS/SOMED records 2010–2016. For SE-participants with several hospitalisations, each MS and SOMED record was linked with the corresponding SE-information. The SIHOS inpatient cohort contains only the matched records of social and inpatient data (N = 987,552) while the SIHOS population cohort contains all SE records 2010–2014 (N = 1.2 million) with indicators for hospitalisations in the two years following SE participation [3].

The SIHOS database underwent comprehensive validation regarding correctness and completeness of the matched records [58]. The validation of the correctness has shown that the extent of mismatches is marginal: only 0.2 percent of MS records and 0.01 percent of SOMED records had to be excluded because age and/or sex did not correspond between MS and SE (Fig 1). Regarding completeness, the validation suggested a matching rate of only 70 percent that could be explained by erroneously built anonymous linkage codes. This resulted in a smaller inpatient cohort than expected, but the missing matches were randomly distributed across the social factors of interest for SIHOS, except for an underrepresentation of non-European migration groups [58].

The current analysis makes use of the SIHOS inpatient cohort that consists of the matched records of the medical data of the MS 2010–2016 with the social data of the SE 2010–2014 (N = 950,182). Included in this study are all records of patients aged 25–84 who were hospitalized for acute care with a main diagnosis (ICD10-GM codes) of one of 15 selected chronic diseases (Table 1), resulting in a sample of N = 141,307 records (Fig 1). The chronic diseases include cancers, diabetes, cardiovascular, respiratory and musculoskeletal diseases and are among the leading chronic conditions in high-income countries according to disability-adjusted life years (DALYs) [59]. They were selected in accordance with the following criteria: (1) chronic condition or an acute incident of a chronic condition (e.g., a myocardial infarction), (2) frequency of the disease in Switzerland, (3) frequency of hospitalisations due to the disease in Switzerland, and (4) percentage of all deaths caused by the disease in Switzerland. Table 1 shows the definition of the 15 selected chronic diseases according to ICD10-GM codes and the Clinical Classification Software (CCS Level 1). The study sample was restricted to these chronic diseases because of their relevance for the health system [59] but also in order to reduce the variability of length of stay related to the high number of different main diagnoses in the inpatient cohort that may not be adequately controlled for in multivariate analysis.

The Northwestern and Central Switzerland Ethics Committee confirmed that the quantitative part of the SIHOS study is exempted from ethics committee approval according to the Swiss Human Research Act, because it is based on anonymized administrative data (2017–01125).

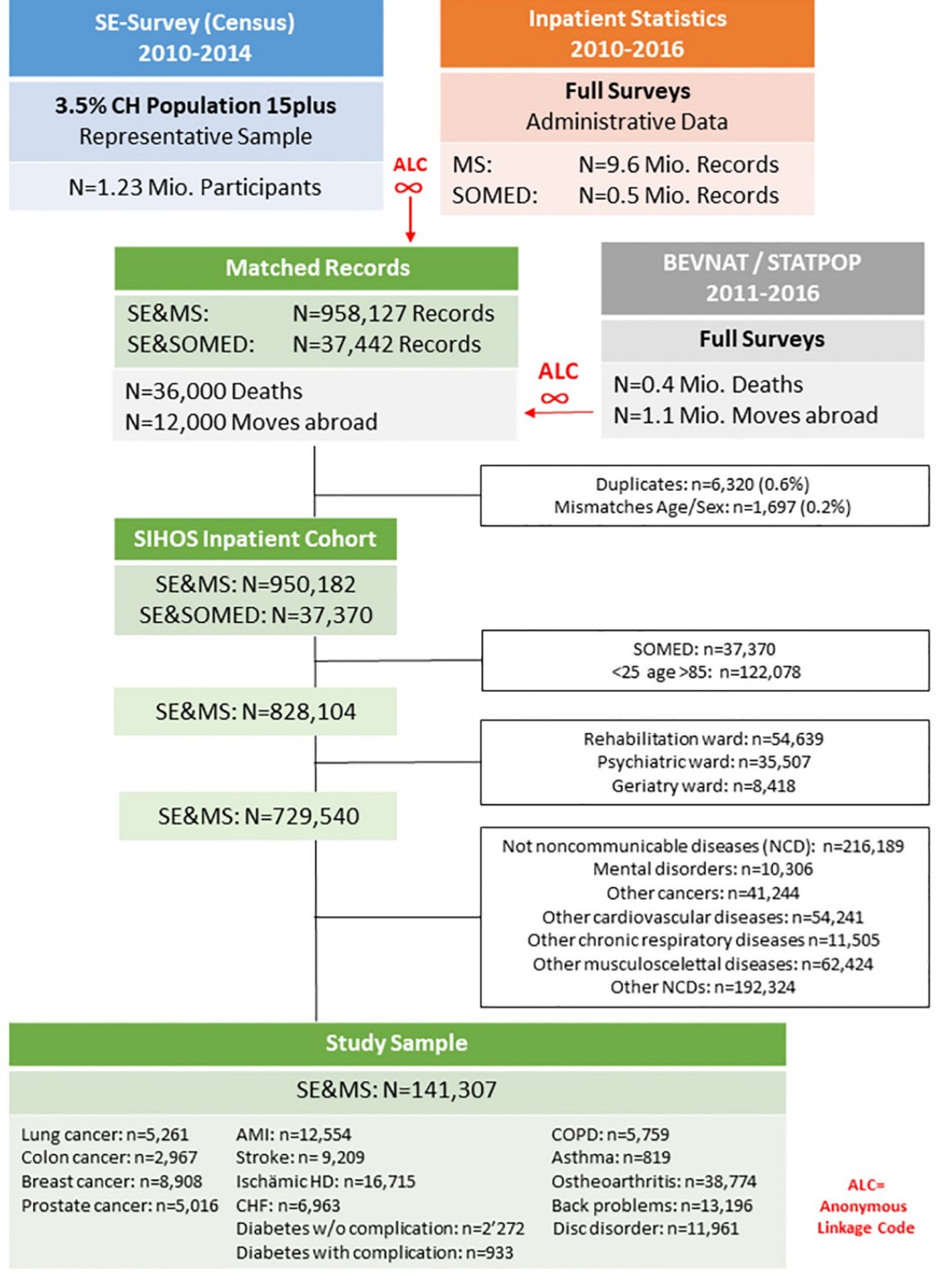

**Fig 1. Flow chart of record matching and selection process for the SIHOS study sample.**

## Definition of variables

**Outcome.** *Length of hospital stay (LOS)* was based on SwissDRG definition, calculated by day of admission and each subsequent day without the day of discharge and excluding days of leave [60].

**Table 1. Definition of specific chronic diseases based on main diagnosis during hospitalisation.**

| Specific chronic diseases (main diagnosis) | CCS Level 1* | ICD10-GM Codes (version 2017) |
|---|---|---|
| Malignant neoplasms (cancer) | | |
| Lung Cancer | CCS_LEV1 = 19 | C34, D022 |
| Colon Cancer | CCS_LEV1 = 14 | C18, D010 |
| Breast Cancer (women) | CCS_LEV1 = 24 | C50, D05 |
| Prostate Cancer (men) | CCS_LEV1 = 29 | C61, D075 |
| Cardiovascular diseases (incl. risk factors) | | |
| Diabetes w/o complications | CCS_LEV1 = 49 | E109, E119, E139, E149, R73 excl. E12 (Diabetes related to Malnutrition) |
| Diabetes with complications | CCS_LEV1 = 50 | E10-E14; 3rd/4th decimal place for complications (excl. 3rd decimal place = 9 = w/o complication) |
| Congestive heart failure (CHF) | CCS_LEV1 = 108 | I50 |
| Ischaemic heart disease | CCS_LEV1 = 101 | I20, I24, I25 |
| Acute myocardial infarction (AMI) | CCS_LEV1 = 100 | I21, I22 |
| Acute cerebrovascular diseases | CCS_LEV1 = 109 | I60-I64, I66 |
| Chronic respiratory diseases | | |
| Chronic obstructive pulmonary disease (COPD) | CCS_LEV1 = 127 | J40-J44, J47 |
| Asthma | CCS_LEV1 = 128 | J45, J46 |
| Musculoskeletal diseases | | |
| Osteoarthritis | CCS_LEV1 = 203 | M15-M19 |
| Back problems | CCS_LEV1 = 205 (excl. ICD10 = M50/51) | M43.2, M43.3, M43.4, M43.5, M43.6, M45, M46 (excl. M46.2, M46.3), M47, M48 (excl. M48.5), M49 (excl. M49.0, M49.5), M53, M54 |
| Disc disorders | N/A | M50, M51 |

*CCS = Clinical Classifications Software; developed by the Healthcare Cost and Utilization Project (HCUP), financed by the US-Agency for Healthcare Research and Quality, adapted for Switzerland by Daniel Zahnd, Bern University of Applied Sciences

**Indicators of health status.** As indicators of the current health status of inpatients, the *main diagnosis* of the hospital stay (one of the 15 specific chronic diseases, Table 1) and information on *inpatient comorbidity* were available. In the SIHOS database, different measures have been defined as indicators of comorbidity. The number of health conditions has been reported to be a simple, yet well performing indicator of multimorbidity for inpatients of medical wards [61] and a recent study suggested that taking into account specific diagnoses did not provide much gain [62]. In the SIHOS inpatient cohort, the number of side diagnoses (NSD, truncated to 13) selected for the current analysis, was linearly associated with length of stay (ANOVA: p<0.001) and proved to better predict length of stay than the number of Elixhauser-VanWalraven Comorbidities [63]. For multivariate analyses, NSD was centred by main diagnosis, allowing to control for different means and within-group variability of NSD [64]. Psychic comorbidity has been found to be related to longer hospital stays in acute hospitalisations in Switzerland [65]. Therefore, the binary variable psychic comorbidity (1 = psychic SD, 0 = no psychic SD) was used as second indicator of inpatient comorbidity.

**Demographic factors.** The *demographic variables* age, sex, and nationality (grouped into (1) Swiss, (2) EU/EFTA and (3) other nationality) were available from the Hospital Medical Statistics. For multivariate analyses, age was implemented with four variables: age centred by main diagnosis, allowing to control for different means and within-group variability of age [64], and three restricted cubic spline-variables, allowing for non-linear associations of age

with the outcome variables of the linear (LOS, NSD) and logistic (transfer to inpatient setting) regression models [66].

**Indicators of social situation.**   As an indicator for *educational attainment*, the SIHOS database includes from the SE the highest educational qualification achieved, grouped into (1) compulsory education, (2) upper secondary level (mainly vocational education) and (3) tertiary level (advanced professional levels and university). This is a meaningful value from around the age of 25 upwards [67]. Educational attainment is a classic indicator of vertical social inequality and displays a strong and consistent relationship with the population's health opportunities and risks of disease and mortality [2, 68].

*Hospital insurance class* is another indicator of vertical social inequality that is available from the Hospital Medical Statistics. It is used as proxy for financial resources since there is no direct information on income in the SIHOS database. However, it may also have an impact on the type and volume of medical interventions. The variable is grouped into three categories: (1) general, mandatory insurance, (2) semi-private insurance and (3) private insurance. The use of insurance class as a proxy for financial resources is supported by a recent study that shows that the Swiss population with private or semi-private hospital insurance has a higher income and a higher level of education compared to the population without this supplementary insurance [69]. There is also evidence that insurance class in its function as a financial incentive system has an impact on the use of health care and type of treatment during the hospital stay [69, 70].

As an indicator for a *person's social resources*, the SIHOS database contains the variable household type from the SE, dichotomized into (1) living with others and (2) living alone. People who live alone in a household have a demonstrably higher risk of receiving less social support and feeling lonelier than people who live with others [18]. Living alone does not preclude a person from having a large, strong social network. Nonetheless, there is a lack of immediate everyday support, which people living in the same household may provide in the event of health problems or after hospital discharge.

The indicator for *migration background*, derived from the SE, distinguishes the following three categories: (1) Swiss national without migration background, (2) second or higher order generation migrant or Swiss national with migration background and (3) first generation migrant (person born abroad). The indicator for *language skills*, also derived from the SE, distinguishes three categories: (1) speaks the local language, (2) does not speak the local language, but another official language or English and (3) allophone, i.e., speaks neither an official language of Switzerland (German, French, Italian and Romansh) nor English. The organization of translation services for allophone patients or for patients who do not have good command of the local language may delay medical examinations and treatment or the organization of discharge.

**Factors related to hospital stay.**   Regarding *treatment in hospital* two variables were derived from variables of the Hospital Medical Statistics: "hospital ward" was dichotomized into (1) surgical ward and (2) internal medicine or other ward. "Need of intensive care" was dichotomized from hours of intensive care into (1) yes, (2) no.

*Discharge destination* was also available from the Hospital Medical Statistics and grouped into three categories: (1) discharge to own home, (2) transfer to another inpatient setting and (3) patient died in hospital. For a sensitivity analysis in the multilevel regression analysis, patients who died in hospital were excluded and a binary discharge variable with the first two categories was used. For the mediation analysis categories 1 and 3 were collapsed and the binary variable *transfer* was defined with (0) not transferred and (1) transfer to another inpatient setting.

On the hospital level, the variable "*language region of the hospital*" is available in the SIHOS database. It distinguishes hospitals located in the (1) German speaking, (2) French speaking, (3) Italian speaking and (4) bilingual parts of Switzerland. For data protection reasons, the SIHOS data set does not contain any geographical information on hospital or individual level but includes a variable from the SE assessing the main language spoken by the patients. Assuming that a hospital in which the majority of patients (70% and more) indicate German (including Romansh), French or Italian as their main language is located in the respective region, most hospitals could be assigned to one of the three main language regions. Seven out of 221 hospitals (3.2%) must have been located in a mixed (bilingual) language region. Hospitals with less than 50 records were excluded from classification and the language region was coded as missing. Finally, the *year of discharge* was available to account for the introduction of SwissDRG in 2012 as well as to control for possible secular trends and was implemented as a categorical variable with seven levels.

## Statistical analysis

Statistical analyses were performed using IBM SPSS Statistics Version 26. Descriptive statistics of the outcome variable (length of stay) and the postulated mediators (number of side diagnosis and transfer to inpatient setting) are reported for the demographic, social, health and hospital related factors and include mean (SD), median (IQR) and percentages, as appropriate. Multilevel models involved the crossed clustering levels hospital and patient (CCMM, Fig 2) reflecting the fact that some patients were hospitalized in different hospitals [54]. Differences between hospitals on the organizational or system level may have an impact on the length of stay. In a first stage (I), linear CCMM were used to investigate the associations between the social factors and the continuous outcome length of stay [71]. In a second stage (II), two mediation analyses for length of stay were conducted, one with the continuous intermediate outcome number of side diagnoses (linear CCMM) and the other with the binary intermediate outcome transfer to inpatient setting (logistic CCMM). In the null model, the ICC for patients was 0.15 (n = 141,307 records and n = 92,623 clusters) and for hospitals 0.28 (n = 141,307 records and n = 188 clusters; S1 Table). Residuals of the linear CCMM were normally distributed, but visual inspection of residuals vs. predicted values plots suggested heteroscedasticity that was confirmed with the modified Breusch-Pagan test (p<0.001). Therefore, robust confidence intervals and p-values are reported for all models (GENLINMIXED procedure of SPSS).

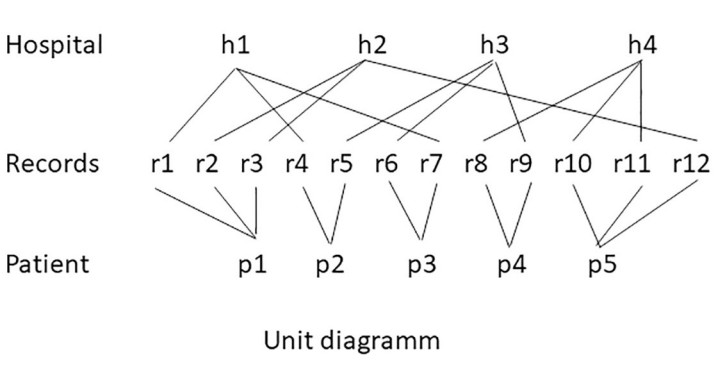 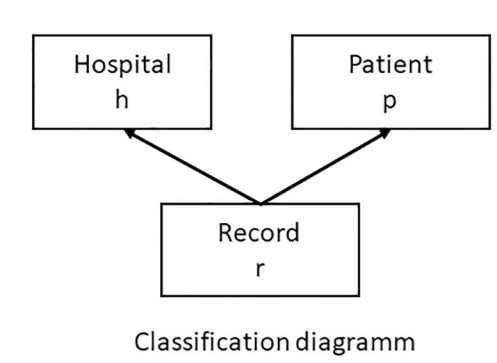

**Fig 2. Cross-classified multilevel data structure.**

In stage (I), fixed effects were introduced in four steps: *Model A* includes the *main predictors* educational attainment, insurance class and household type and controls for demography (sex, age, and nationality), main diagnosis (one of the selected 15 chronic diseases), language region of hospital and year of discharge. *Model B* introduces the postulated mediator number of side diagnoses and psychic comorbidity. *Model C* additionally controls for the factors related to the hospital stay (hospital ward and need of intensive care) and lastly, *Model D* includes the postulated mediator discharge destination. Records with missing values in one of the variables included in Model D (n = 404 records) were excluded in all CCMM, resulting in a final sample size of n = 140,903 records.

To assess the impact of *migration factors* on length of stay, Model D was performed excluding nationality and including either language skills (Model D.1) or migration status (Model D.2), because multicollinearity was observed between nationality, migration status and language skills (Cramer's V nationality vs. language skills: 0.425, p<0.001; nationality vs. migration status: 0.689, p<0.001; language skills vs. migration status: 0.402, p<0.001). No indication for multicollinearity was found between the other social factors, demography, and the main diagnosis (Cramer's V <0.2), except for sex vs. main diagnosis (Cramer's V: 0.403, p<0.001). This could be explained by the fact that only men can have a main diagnosis of prostate cancer and that in our study sample only woman have a main diagnosis of breast cancer. Taking into account the interaction between sex and main diagnoses marginally changed the effect estimates of sex for the other specific chronic diseases, but not those of the main predictors and the other covariates in Model D.

For the fully adjusted Model D, *statistical interaction* was tested between all social factors, between sex and social factors and between main diagnoses and social factors by introducing the corresponding two-way interaction terms one by one (Akaike criteria for model improvement). Conditional effects rather than stratified estimates are reported for significant interactions [72].

To test and quantify the postulated *mediating effect of comorbidity* on the association between educational attainment and length of stay, a linear CCMM was used in stage II with number of side diagnoses as continuous intermediate outcome and educational attainment as predictor while controlling for the covariates of Model A. For the indirect effect of educational attainment on length of stay, Monte Carlo confidence intervals were computed with the macro mcmed of PROCESS [72]. In a similar way mediation of the association between household type and length of stay by number of side diagnosis was evaluated.

The postulated *mediating effect of discharge destination* on the association between household type and length of stay was evaluated with a logistic CCMM, with the binary intermediate outcome transfer (0 = not transferred and 1 = transfer to another inpatient setting) and the main predictor household type, while controlling for the covariates of Model C. The significance of the indirect effect of household type on length of stay, which combines effect estimates of linear and logistic regression, was assessed according to Iacobucci [73].

## Results

Characteristics of the study population are presented in Table 2 (demographic and social factors) and Table 3 (factors related to health status and hospital stay). The mean length of stay was 7.00 (SD 6.50) days and varied between 3.70 (5.93) days for patients hospitalized for ischaemic heart disease and 11.70 (10.25) days for patients with colon cancer. The number of side diagnoses (mean 3.61, SD 3.26) varied between 1.98 (2.40) in patients with disc disorders and 7.69 (3.31) in patients with congestive heart failure. 22% of inpatients were transferred to another inpatient setting, varying between 49% (stroke) and 2% (breast cancer). Fig 3, top,

**Table 2. Distribution (N records, %) of demographic and social factors with descriptive statistics (mean (SD), median (IQR)) of length of stay and number of side diagnoses and percentage (%) of transfer to inpatient setting = yes.**

| | Records | | Length of stay | | | | | N° Side diagnoses | | | | | Transfer to inpatient setting[a] |
|---|---|---|---|---|---|---|---|---|---|---|---|---|---|
| | N | % | Mean (days) | SD (days) | Median | IQR | | Mean (N) | SD (N) | Median | IQR | | Yes (%) |
| **Total** | 141'307 | 100% | 7.00 | 6.50 | 6 | 3 | 9 | 3.61 | 3.26 | 3 | 1 | 5 | 22% |
| **Age** | | | | | | | | | | | | | |
| 25–44 years | 9'282 | 6.6% | 5.14 | 4.93 | 4 | 2 | 6 | 1.75 | 2.19 | 1 | 0 | 3 | 10% |
| 45–64 years | 50'203 | 35.5% | 6.14 | 6.09 | 5 | 2 | 8 | 2.82 | 2.81 | 2 | 1 | 4 | 15% |
| 65–84 years | 81'822 | 57.9% | 7.73 | 6.79 | 7 | 3 | 10 | 4.30 | 3.42 | 4 | 2 | 6 | 27% |
| **Sex** | | | | | | | | | | | | | |
| Men | 77'525 | 54.9% | 6.78 | 6.71 | 5 | 2 | 9 | 3.87 | 3.35 | 3 | 1 | 6 | 20% |
| Women | 63'782 | 45.1% | 7.27 | 6.22 | 6 | 3 | 9 | 3.29 | 3.12 | 2 | 1 | 5 | 23% |
| **Nationality** | | | | | | | | | | | | | |
| Swiss | 122'813 | 86.9% | 6.99 | 6.45 | 6 | 3 | 9 | 3.58 | 3.25 | 3 | 1 | 5 | 22% |
| EU/EFTA | 13'604 | 9.6% | 7.21 | 6.98 | 6 | 3 | 9 | 3.76 | 3.34 | 3 | 1 | 6 | 22% |
| Other nationality | 4'885 | 3.5% | 6.60 | 6.38 | 5 | 2 | 9 | 3.83 | 3.30 | 3 | 1 | 6 | 18% |
| Missing values | 5 | 0.0% | 7.20 | 3.83 | 8 | 5 | 9 | 4.60 | 4.88 | 4 | 0 | 8 | 20% |
| **Educational attainment** | | | | | | | | | | | | | |
| Compulsory | 41'082 | 29.1% | 7.59 | 6.60 | 6 | 3 | 10 | 4.05 | 3.37 | 3 | 1 | 6 | 25% |
| Upper secondary | 71'601 | 50.7% | 6.89 | 6.53 | 6 | 3 | 9 | 3.53 | 3.23 | 3 | 1 | 5 | 21% |
| Tertiary | 28'624 | 20.3% | 6.42 | 6.21 | 5 | 2 | 8 | 3.15 | 3.08 | 2 | 1 | 5 | 18% |
| **Insurance class** | | | | | | | | | | | | | |
| Mandatory | 96'132 | 68.0% | 6.98 | 6.61 | 6 | 3 | 9 | 3.77 | 3.33 | 3 | 1 | 6 | 23% |
| Semi-private | 29'641 | 21.0% | 6.88 | 6.04 | 6 | 3 | 9 | 3.27 | 3.09 | 2 | 1 | 5 | 18% |
| Private | 15'524 | 11.0% | 7.30 | 6.65 | 6 | 3 | 10 | 3.22 | 3.06 | 2 | 1 | 5 | 18% |
| Missing values | 10 | 0.0% | 4.80 | 3.46 | 4 | 3 | 5 | 2.00 | 3.09 | 1 | 0 | 2 | 20% |
| **Type of household** | | | | | | | | | | | | | |
| Living with others | 102'905 | 72.8% | 6.66 | 6.18 | 5 | 3 | 9 | 3.45 | 3.20 | 3 | 1 | 5 | 18% |
| Living alone | 38'402 | 27.2% | 7.91 | 7.21 | 7 | 3 | 10 | 4.01 | 3.39 | 3 | 1 | 6 | 30% |
| **Language skills** | | | | | | | | | | | | | |
| At least regional language | 123'322 | 87.3% | 7.00 | 6.49 | 6 | 3 | 9 | 3.58 | 3.25 | 3 | 1 | 5 | 21% |
| At least one official language or English | 10'358 | 7.3% | 6.93 | 6.51 | 6 | 2 | 9 | 3.71 | 3.29 | 3 | 1 | 5 | 24% |
| No official language and no English | 5'937 | 4.2% | 6.99 | 6.96 | 5 | 2 | 9 | 3.87 | 3.34 | 3 | 1 | 6 | 19% |
| Missing values | 1'690 | 1.2% | 7.06 | 5.88 | 6 | 3 | 9 | 3.91 | 3.43 | 3 | 1 | 6 | 19% |
| **Migration status** | | | | | | | | | | | | | |
| Swiss w/o migration background | 103'511 | 73.3% | 6.97 | 6.41 | 6 | 3 | 9 | 3.58 | 3.26 | 3 | 1 | 6 | 22% |
| 2nd or higher generation migrant | 19'065 | 13.5% | 7.05 | 6.65 | 6 | 3 | 9 | 3.49 | 3.18 | 3 | 1 | 5 | 22% |
| 1st generation migrant | 17'265 | 12.2% | 7.09 | 6.82 | 6 | 2 | 9 | 3.82 | 3.34 | 3 | 1 | 5 | 21% |
| Missing values | 1'466 | 1.0% | 7.31 | 6.74 | 6 | 3 | 9 | 4.05 | 3.31 | 3 | 1 | 6 | 23% |

[a]n = 197 missing values

illustrates for both mean length of stay and mean comorbidity, the almost linear increase with age as well as the social gradient by educational attainment. Fig 3, bottom, shows that those living alone stayed longer in hospital on average and had a higher probability for transfer to another inpatient setting compared to those living with others (unadjusted estimates). All unadjusted associations between length of stay, number of side diagnoses and probability of transfer by age and the social factors are documented in S1–S3 Figs.

**Table 3.** Distribution (N records, %) of variables related to health status and hospital stay with descriptive statistics (mean (SD), median (IQR)) of length of stay and number of side diagnoses and percentage (%) of transfer to inpatient setting = yes.

| | Records | | Length of stay | | | | | N° Side diagnoses | | | | | Transfer to inpatient setting[a] |
|---|---|---|---|---|---|---|---|---|---|---|---|---|---|
| | N | % | Mean (days) | SD (days) | Median | IQR | | Mean (N) | SD (N) | Median | IQR | | Yes (%) |
| Total | 141'307 | 100.0% | 7.00 | 6.50 | 6 | 3 | 9 | 3.61 | 3.26 | 3 | 1 | 5 | 22% |
| **Main diagnosis** | | | | | | | | | | | | | |
| Lung cancer | 5'261 | 3.7% | 8.74 | 9.32 | 6 | 2 | 12 | 5.46 | 3.57 | 5 | 3 | 8 | 17% |
| Colon cancer | 2'967 | 2.1% | 11.70 | 10.25 | 9 | 6 | 15 | 5.04 | 3.66 | 4 | 2 | 8 | 14% |
| Breast cancer | 8'908 | 6.3% | 4.64 | 4.02 | 4 | 2 | 6 | 2.08 | 2.45 | 1 | 0 | 3 | 2% |
| Prostate cancer | 5'016 | 3.5% | 6.56 | 4.98 | 6 | 4 | 8 | 2.73 | 2.85 | 2 | 1 | 4 | 3% |
| Diabetes w/o complications | 933 | 0.7% | 7.14 | 6.22 | 6 | 4 | 9 | 4.05 | 2.82 | 4 | 2 | 6 | 9% |
| Diabetes with complications | 2'272 | 1.6% | 11.06 | 12.45 | 7 | 3 | 13 | 6.23 | 3.58 | 6 | 3 | 9 | 16% |
| Acute myocardial infarction | 12'554 | 8.9% | 5.41 | 6.44 | 4 | 1 | 7 | 4.65 | 3.17 | 4 | 2 | 6 | 42% |
| Acute cerebrovascular disease | 9'209 | 6.5% | 8.78 | 8.14 | 7 | 4 | 12 | 5.53 | 3.44 | 5 | 3 | 8 | 49% |
| Ischaemic heart disease | 16'715 | 11.8% | 3.70 | 5.93 | 1 | 1 | 3 | 3.88 | 2.82 | 3 | 2 | 5 | 15% |
| Congestive heart failure | 6'963 | 4.9% | 9.92 | 8.73 | 8 | 4 | 13 | 7.69 | 3.31 | 8 | 5 | 10 | 22% |
| COPD | 5'759 | 4.1% | 8.66 | 7.07 | 7 | 4 | 11 | 5.29 | 3.33 | 5 | 3 | 8 | 24% |
| Asthma | 819 | 0.6% | 5.13 | 4.58 | 4 | 2 | 7 | 3.16 | 2.87 | 2 | 1 | 5 | 10% |
| Osteoarthritis | 38'774 | 27.4% | 7.49 | 4.22 | 7 | 5 | 9 | 2.19 | 2.29 | 2 | 0 | 3 | 24% |
| Back Problems | 13'196 | 9.3% | 7.68 | 6.38 | 6 | 4 | 9 | 3.49 | 3.04 | 3 | 1 | 5 | 18% |
| Disc disorder | 11'961 | 8.5% | 6.41 | 4.65 | 5 | 4 | 8 | 1.98 | 2.40 | 1 | 0 | 3 | 10% |
| **Number of side diagnosis** | | | | | | | | | | | | | |
| None | 23'885 | 16.9% | 5.64 | 4.55 | 5 | 3 | 8 | 0.00 | 0.00 | 0 | 0 | 0 | 14% |
| 1 | 21'915 | 15.5% | 5.27 | 3.80 | 5 | 2 | 7 | 1.00 | 0.00 | 1 | 1 | 1 | 15% |
| 2 | 19'998 | 14.2% | 5.56 | 4.16 | 5 | 2 | 8 | 2.00 | 0.00 | 2 | 2 | 2 | 17% |
| 3 | 17'078 | 12.1% | 5.90 | 4.54 | 5 | 2 | 8 | 3.00 | 0.00 | 3 | 3 | 3 | 20% |
| 4 | 13'719 | 9.7% | 6.38 | 5.04 | 6 | 2 | 9 | 4.00 | 0.00 | 4 | 4 | 4 | 22% |
| 5 | 10'750 | 7.6% | 6.89 | 5.29 | 6 | 3 | 9 | 5.00 | 0.00 | 5 | 5 | 5 | 24% |
| 6 | 8'128 | 5.8% | 7.71 | 6.17 | 7 | 3 | 10 | 6.00 | 0.00 | 6 | 6 | 6 | 26% |
| 7 | 6'194 | 4.4% | 8.48 | 6.43 | 7 | 4 | 11 | 7.00 | 0.00 | 7 | 7 | 7 | 30% |
| 8 | 4'618 | 3.3% | 9.33 | 7.05 | 8 | 4 | 12 | 8.00 | 0.00 | 8 | 8 | 8 | 31% |
| 9 | 4'777 | 3.4% | 11.81 | 10.74 | 9 | 6 | 15 | 9.00 | 0.00 | 9 | 9 | 9 | 35% |
| 10 | 3'212 | 2.3% | 12.48 | 10.38 | 10 | 6 | 15 | 10.00 | 0.00 | 10 | 10 | 10 | 37% |
| 11 | 2'432 | 1.7% | 13.92 | 11.34 | 11 | 7 | 17 | 11.00 | 0.00 | 11 | 11 | 11 | 41% |
| 12 | 3'097 | 2.2% | 16.24 | 13.03 | 13 | 8 | 20 | 12.00 | 0.00 | 12 | 12 | 12 | 45% |
| 13 or more | 1'504 | 1.1% | 17.37 | 14.60 | 14 | 8 | 22 | 13.00 | 0.00 | 13 | 13 | 13 | 47% |
| **Psychic comorbidity** | | | | | | | | | | | | | |
| No | 130'554 | 92.4% | 6.76 | 6.28 | 6 | 3 | 9 | 3.37 | 3.16 | 3 | 1 | 5 | 20% |
| Yes | 10'753 | 7.6% | 9.83 | 8.29 | 8 | 5 | 13 | 6.41 | 3.17 | 6 | 4 | 9 | 34% |
| **Hospital ward** | | | | | | | | | | | | | |
| Internal medicine or other ward | 71'159 | 50.4% | 6.39 | 7.10 | 4 | 2 | 8 | 4.56 | 3.43 | 4 | 2 | 7 | 22% |
| Surgical ward | 70'148 | 49.6% | 7.62 | 5.76 | 7 | 5 | 9 | 2.64 | 2.76 | 2 | 1 | 4 | 21% |
| Intensive care | | | | | | | | | | | | | |
| No need of intensive care | 125'513 | 88.8% | 6.59 | 5.72 | 6 | 3 | 9 | 3.30 | 3.09 | 2 | 1 | 5 | 19% |
| Yes, need for intensive care | 15'623 | 11.1% | 10.24 | 10.35 | 8 | 4 | 13 | 6.02 | 3.55 | 6 | 3 | 9 | 42% |
| Missing Values | 171 | 0.1% | 7.96 | 6.24 | 6 | 4 | 10 | 4.36 | 3.08 | 4 | 2 | 6 | 27% |
| Discharge destination | | | | | | | | | | | | | |
| Discharge to home | 108'557 | 76.8% | 6.23 | 5.33 | 5 | 3 | 8 | 3.19 | 2.98 | 2 | 1 | 5 | 0% |

*(Continued)*

**Table 3.** (Continued)

| | Records | | Length of stay | | | | | N° Side diagnoses | | | | | Transfer to inpatient setting[a] |
|---|---|---|---|---|---|---|---|---|---|---|---|---|---|
| | N | % | Mean (days) | SD (days) | Median | IQR | | Mean (N) | SD (N) | Median | IQR | | Yes (%) |
| Transfer to inpatient setting | 30'354 | 21.5% | 9.53 | 8.73 | 8 | 4 | 12 | 4.82 | 3.68 | 4 | 2 | 7 | 100% |
| Died in hospital | 2'199 | 1.6% | 9.95 | 11.47 | 6 | 2 | 14 | 7.42 | 3.76 | 8 | 4 | 11 | 0% |
| Missing Values | 197 | 0.1% | 7.02 | 4.65 | 6 | 4 | 10 | 2.29 | 2.81 | 1 | 0 | 4 | 77% |
| Hospital language region | | | | | | | | | | | | | |
| German | 94'584 | 66.9% | 6.79 | 6.19 | 6 | 3 | 9 | 3.70 | 3.26 | 3 | 1 | 6 | 20% |
| French | 33'697 | 23.8% | 7.39 | 7.02 | 6 | 3 | 9 | 3.38 | 3.31 | 3 | 1 | 5 | 25% |
| Italian | 11'288 | 8.0% | 7.53 | 7.41 | 6 | 2 | 10 | 3.44 | 3.05 | 2 | 1 | 5 | 20% |
| Mixed | 1'717 | 1.2% | 7.00 | 5.88 | 6 | 3 | 9 | 3.98 | 3.43 | 3 | 1 | 5 | 19% |
| Missing Values | 21 | 0.0% | 11.90 | 4.52 | 14 | 11 | 14 | 0.05 | 0.22 | 0 | 0 | 0 | 5% |
| Discharge year | | | | | | | | | | | | | |
| 2010 | 16'801 | 11.9% | 7.05 | 6.33 | 6 | 3 | 9 | 2.44 | 2.44 | 2 | 0 | 4 | 21% |
| 2011 | 18'269 | 12.9% | 7.10 | 6.40 | 6 | 3 | 9 | 2.73 | 2.58 | 2 | 1 | 4 | 21% |
| 2012 | 18'766 | 13.3% | 7.20 | 6.35 | 6 | 3 | 9 | 3.30 | 2.90 | 3 | 1 | 5 | 21% |
| 2013 | 20'608 | 14.6% | 6.90 | 6.40 | 6 | 3 | 9 | 3.60 | 3.14 | 3 | 1 | 5 | 21% |
| 2014 | 21'309 | 15.1% | 7.10 | 6.95 | 6 | 3 | 9 | 4.02 | 3.42 | 3 | 1 | 6 | 22% |
| 2015 | 21'182 | 15.0% | 6.90 | 6.33 | 6 | 3 | 9 | 4.21 | 3.53 | 3 | 1 | 6 | 22% |
| 2016 | 24'372 | 17.2% | 6.81 | 6.63 | 5 | 3 | 8 | 4.41 | 3.74 | 3 | 1 | 7 | 23% |

[a]n = 197 missing values

Results of the multilevel linear regression models (Models A-D) are presented in Table 4. *Educational attainment.* According to Model A, the average length of hospital stay was increased by 0.24 and 0.37 days among patients with upper secondary education and compulsory education, respectively, compared to patients with tertiary education (both p<0.001). After adjustment for the two indicators of comorbidity (NSD and psychic comorbidity) the differences collapsed and were no longer significant (Model B). The inclusion of factors related to treatment (Model C) and discharge (Model D) did not further change effect estimates. The tests for interaction in Model D showed evidence for an interaction between educational attainment and main diagnosis (S2 Table). Colon cancer, COPD, asthma and ischaemic heart disease showed significant effects also in the fully adjusted model: Patients with compulsory education stayed significantly longer in hospital compared to those with tertiary education if they had the main diagnosis colon cancer (0.99 days, 95% CI: 0.13, 1.84; p = 0.024) or asthma (0.93, 0.06, 1.80; p = 0.036) and those with upper secondary education stayed significantly longer with the main diagnosis COPD (0.56; 0.02, 1.10, p = 0.041) while ischaemic heart disease patients with upper secondary education had somewhat shorter stays (-0.17;-0.33, -0.01, p = 0.039).

## Hospital insurance class

According to Model A, patients with private insurance stayed on average 0.30 days longer compared to patients with basic insurance (p = 0.001). Inclusion of comorbidity and of factors related to treatment and discharge increased the effect size incrementally to 0.36 days (p <0.0001). Semi-private insurance showed a significant effect only after adjustment for hospital-stay related factors and was associated in Model D with 0.15 days longer stays compared to basic insurance (p = 0.028). The tests of interaction in the fully adjusted Model D revealed

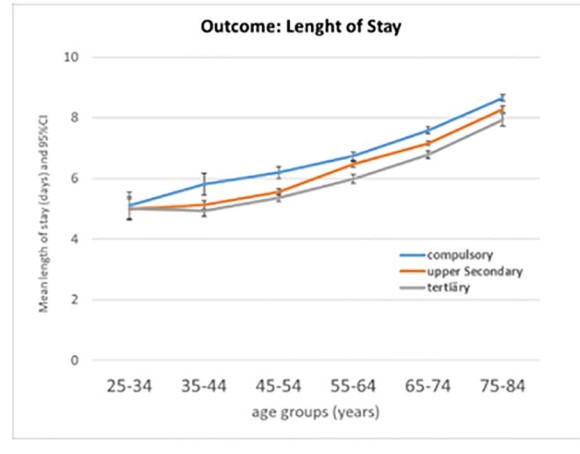
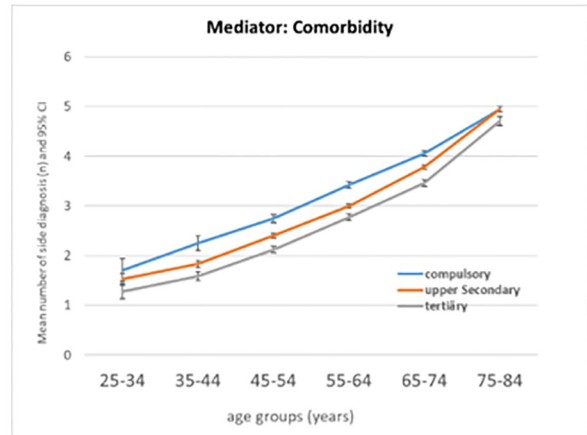
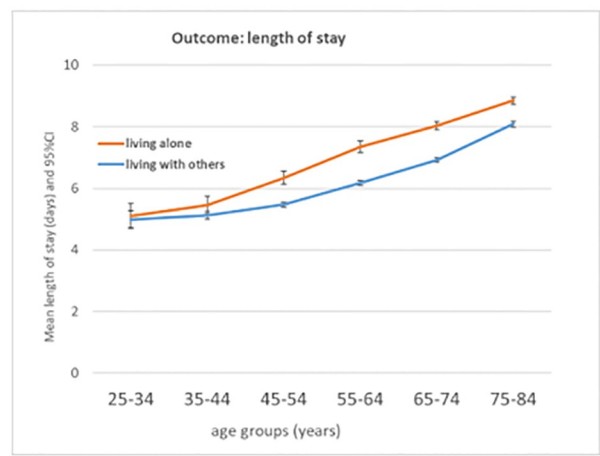
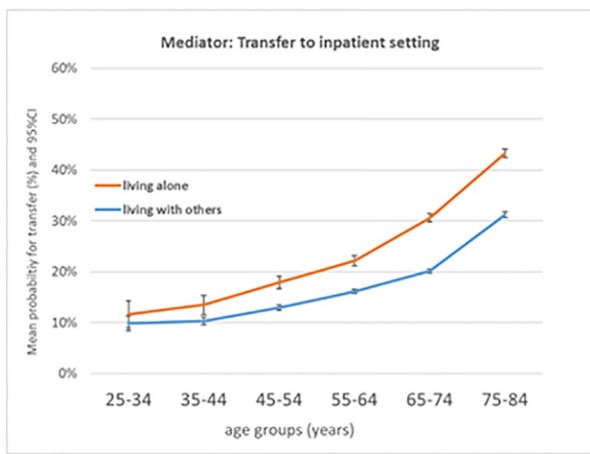

**Fig 3.** Top. Outcome length of stay (left) and mediator comorbidity (right) by age groups and educational attainment; bottom: Outcome length of stay (left) and mediator discharge destination (right) by age groups and household type.

evidence of a moderation effect of the main diagnosis on the association between insurance class and length of stay (S2 Table). Compared to other chronic conditions, stronger effects were found for patients with AMI, COPD, and back problems (patients with private or semi-private insurance stayed on average between 0.54 and 1.50 days longer compared to those with basic insurance), whereas an opposite effect was observed among patients with colon or breast cancer: patients with private or semi-private insurance left the hospital 0.36 to 1.10 days earlier compared to those with basic insurance.

## Household type

In Model A, persons living alone stayed 0.60 days longer in hospital compared to those living with others (p<0.001). Adjustment for comorbidity reduced the effect by about one third to 0.39 days (p<0.001; Modell B) while additional introduction of hospital ward and intensive care (Model C) changed the estimate only slightly. Further adjustment for discharge destination reduced the effect by about another third to 0.28 days (p<0.01; Model D). The tests for interaction in Model D (S2 Table) showed larger differences of living alone compared to living with others for patients with lung cancer (0.92 days, 95% CI: 0.46, 1.38; p<0.001), colon cancer (1.21, 0.49, 1.94; p = 0.001) and back problems (0.56, 0.30, 0.82; p<0.001).

**Table 4. Associations of length of stay with social factors, health status and factors related to hospital stay (linear CCMM A to D).**

| Outcome: LOS (days) | Model A (N = 140'903) | | | | Model B (N = 140'903) | | | | Model C (N = 140'903) | | | | Model D (N = 140'903) | | | |
|---|---|---|---|---|---|---|---|---|---|---|---|---|---|---|---|---|
| Fixed effects[a] | β (days) | p-value | 95% CI | | β (days) | p-value | 95% CI | | β (days) | p-value | 95% CI | | β (days) | p-value | 95% CI | |
| | | | Lower | Upper | | | Lower | Upper | | | Lower | Upper | | | Lower | Upper |
| Intercept | 2.47 | 0.43 | -3.63 | 8.56 | 0.17 | 0.96 | -6.19 | 6.53 | -1.76 | 0.58 | -8.10 | 4.57 | -2.06 | 0.53 | -8.49 | 4.36 |
| **Educational attainment** | | | | | | | | | | | | | | | | |
| Compulsory | 0.37 | <0.001 | 0.27 | 0.47 | 0.03 | 0.55 | -0.07 | 0.12 | 0.05 | 0.29 | -0.04 | 0.14 | 0.05 | 0.29 | -0.04 | 0.14 |
| Upper secondary | 0.24 | <0.001 | 0.14 | 0.33 | 0.03 | 0.53 | -0.06 | 0.12 | 0.03 | 0.47 | -0.05 | 0.11 | 0.04 | 0.38 | -0.04 | 0.12 |
| Tertiary | Ref. | | | | Ref. | | | | Ref. | | | | Ref. | | | |
| **Insurance class** | | | | | | | | | | | | | | | | |
| Private | 0.30 | <0.001 | 0.13 | 0.47 | 0.30 | <0.001 | 0.14 | 0.46 | 0.33 | <0.001 | 0.16 | 0.50 | 0.36 | <0.001 | 0.18 | 0.53 |
| Semi-private | 0.06 | 0.40 | -0.08 | 0.19 | 0.10 | 0.11 | -0.02 | 0.23 | 0.13 | 0.04 | 0.00 | 0.26 | 0.15 | 0.03 | 0.02 | 0.29 |
| Mandatory | Ref. | | | | Ref. | | | | Ref. | | | | Ref. | | | |
| **Household type** | | | | | | | | | | | | | | | | |
| Living alone | 0.60 | <0.001 | 0.50 | 0.70 | 0.39 | <0.001 | 0.30 | 0.48 | 0.42 | <0.001 | 0.34 | 0.51 | 0.28 | <0.001 | 0.20 | 0.37 |
| Living with others | Ref. | | | | Ref. | | | | Ref. | | | | Ref. | | | |
| **Sex** | | | | | | | | | | | | | | | | |
| Men | -0.15 | <0.001 | -0.25 | -0.05 | -0.35 | <0.001 | -0.43 | -0.26 | -0.39 | <0.001 | -0.48 | -0.31 | -0.30 | <0.001 | -0.38 | -0.22 |
| Women | Ref. | | | | Ref. | | | | Ref. | | | | Ref. | | | |
| **Nationality** | | | | | | | | | | | | | | | | |
| Other nationality | 0.36 | <0.001 | 0.13 | 0.58 | 0.07 | 0.49 | -0.14 | 0.28 | 0.25 | 0.02 | 0.04 | 0.45 | 0.22 | 0.03 | 0.02 | 0.42 |
| EU/EFTA | 0.16 | 0.02 | 0.03 | 0.29 | 0.14 | 0.03 | 0.02 | 0.27 | 0.16 | 0.01 | 0.04 | 0.28 | 0.15 | 0.02 | 0.03 | 0.27 |
| Swiss | Ref. | | | | Ref. | | | | Ref. | | | | Ref. | | | |
| **Chronic condition (CC)** | | | | | | | | | | | | | | | | |
| Lung cancer | 4.93 | <0.001 | 4.34 | 5.51 | 5.18 | <0.001 | 4.60 | 5.76 | 4.96 | <0.001 | 4.41 | 5.52 | 4.96 | <0.001 | 4.42 | 5.50 |
| Colon cancer | 8.11 | <0.001 | 7.36 | 8.86 | 8.13 | <0.001 | 7.50 | 8.76 | 6.80 | <0.001 | 6.12 | 7.48 | 6.94 | <0.001 | 6.24 | 7.65 |
| Breast cancer | 1.30 | 0.02 | 0.24 | 2.36 | 1.37 | 0.01 | 0.27 | 2.47 | 1.98 | <0.001 | 0.88 | 3.09 | 2.26 | <0.001 | 1.12 | 3.39 |
| Prostate cancer | 3.27 | <0.001 | 2.73 | 3.82 | 3.29 | <0.001 | 2.88 | 3.71 | 2.19 | <0.001 | 1.61 | 2.77 | 2.48 | <0.001 | 1.88 | 3.08 |
| Diabetes w/o compl. | 3.65 | <0.001 | 2.38 | 4.92 | 3.92 | <0.001 | 2.67 | 5.17 | 4.47 | <0.001 | 3.17 | 5.78 | 4.64 | <0.001 | 3.32 | 5.96 |
| Diabetes with compl. | 7.53 | <0.001 | 6.65 | 8.41 | 7.82 | <0.001 | 6.97 | 8.67 | 7.61 | <0.001 | 6.77 | 8.44 | 7.69 | <0.001 | 6.87 | 8.51 |
| AMI | 1.87 | <0.001 | 1.39 | 2.35 | 2.07 | <0.001 | 1.64 | 2.50 | 1.38 | <0.001 | 0.93 | 1.83 | 0.94 | <0.001 | 0.47 | 1.41 |
| Stroke | 4.81 | <0.001 | 4.01 | 5.60 | 4.93 | <0.001 | 4.36 | 5.49 | 4.42 | <0.001 | 3.95 | 4.90 | 3.87 | <0.001 | 3.37 | 4.37 |
| CHD | 6.00 | <0.001 | 4.96 | 7.04 | 5.94 | <0.001 | 4.98 | 6.89 | 5.84 | <0.001 | 4.93 | 6.76 | 5.83 | <0.001 | 4.91 | 6.75 |
| COPD | 4.72 | <0.001 | 4.01 | 5.44 | 4.78 | <0.001 | 4.14 | 5.41 | 4.96 | <0.001 | 4.42 | 5.50 | 4.90 | <0.001 | 4.31 | 5.49 |
| Asthma | 1.67 | 0.02 | 0.29 | 3.05 | 2.13 | <0.001 | 0.69 | 3.58 | 2.77 | <0.001 | 1.29 | 4.26 | 2.98 | <0.001 | 1.49 | 4.47 |
| Osteoarthritis | 4.18 | <0.001 | 3.70 | 4.66 | 4.21 | <0.001 | 3.85 | 4.58 | 3.15 | <0.001 | 2.61 | 3.69 | 3.06 | <0.001 | 2.51 | 3.61 |
| Back problems | 4.09 | <0.001 | 3.48 | 4.71 | 4.24 | <0.001 | 3.78 | 4.70 | 3.70 | <0.001 | 3.16 | 4.24 | 3.72 | <0.001 | 3.15 | 4.29 |
| Disc disorder | 3.44 | <0.001 | 1.64 | 5.23 | 3.93 | <0.001 | 2.08 | 5.78 | 3.58 | <0.001 | 1.70 | 5.47 | 3.71 | <0.001 | 1.78 | 5.64 |
| Ischaemic heart disease | Ref. | | | | Ref. | | | | Ref. | | | | Ref. | | | |
| **Comorbidity** | | | | | | | | | | | | | | | | |
| NSD (centred by CC) | | | | | 0.89 | <0.001 | 0.80 | 0.98 | 0.83 | <0.001 | 0.74 | 0.92 | 0.80 | <0.001 | 0.70 | 0.89 |
| Psychic comorbidity: yes | | | | | 0.39 | <0.001 | 0.17 | 0.61 | 0.48 | <0.001 | 0.27 | 0.70 | 0.36 | <0.001 | 0.15 | 0.57 |
| Psychic comorbidity: no | | | | | Ref. | | | | Ref. | | | | Ref. | | | |
| **Hospital Ward** | | | | | | | | | | | | | | | | |
| Surgical | | | | | | | | | 1.84 | <0.001 | 1.34 | 2.34 | 1.72 | <0.001 | 1.24 | 2.21 |

*(Continued)*

**Table 4.** (Continued)

| Outcome: LOS (days) | Model A (N = 140'903) | | | | Model B (N = 140'903) | | | | Model C (N = 140'903) | | | | Model D (N = 140'903) | | | |
|---|---|---|---|---|---|---|---|---|---|---|---|---|---|---|---|---|
| Fixed effects[a] | β (days) | p-value | 95% CI | | β (days) | p-value | 95% CI | | β (days) | p-value | 95% CI | | β (days) | p-value | 95% CI | |
| | | | Lower | Upper | | | Lower | Upper | | | Lower | Upper | | | Lower | Upper |
| Internal medicine or other | | | | | | | | | Ref. | | | | Ref. | | | |
| **Need of intensive care** | | | | | | | | | | | | | | | | |
| Yes | | | | | | | | | 3.37 | <0.001 | 2.98 | 3.77 | 3.15 | <0.001 | 2.78 | 3.53 |
| No | | | | | | | | | Ref. | | | | Ref. | | | |
| **Discharge destination** | | | | | | | | | | | | | | | | |
| Died in hospital | | | | | | | | | | | | | 0.34 | 0.31 | -0.32 | 1.01 |
| Transfer to inpatient setting | | | | | | | | | | | | | 1.99 | <0.001 | 1.76 | 2.22 |
| Discharge to home | | | | | | | | | | | | | Ref. | | | |
| Akaike criterion, corrected | 902'403 | | | | 878'281 | | | | 871'531 | | | | 868'810 | | | |

The regression coefficients β are the estimated differences in average length of stay between the respective category and the reference category obtained from the respective model.

[a]All models control for clustering on hospital- and on patient-level and are adjusted for age, language region of hospital and year of discharge

## Migration factors

Non-Swiss nationals stayed about 0.2 days longer in hospital compared to Swiss nationals (Model D) while allophone patients stayed about 0.34 days longer in hospital compared to those who spoke the local language (p = 0.002, Model D.1) (Table 5). Interaction tests in

**Table 5. Associations of length of stay with migration factors (linear CCMM D.1 and D.2).**

| Outcome: LOS (days) | Migration factors instead of nationality | | | |
|---|---|---|---|---|
| Fixed effects[a] | β (days) | p-value | 95% CI | |
| | | | Lower | Upper |
| **Model D.1** (N = 139'191) | | | | |
| **Language skills** | | | | |
| No official language and no English | 0.34 | <0.001 | 0.13 | 0.55 |
| At least one official language or English | 0.00 | 0.98 | -0.12 | 0.13 |
| At least regional language | Ref. | | | |
| **Model D.2** (N = 139'444) | | | | |
| **Migration status** | | | | |
| 1st generation | 0.16 | <0.001 | 0.05 | 0.27 |
| 2nd or higher generation | 0.02 | 0.68 | -0.07 | 0.10 |
| Swiss w/o migration | Ref. | | | |

The regression coefficients β are the estimated differences in average length of stay between the respective category and the reference category obtained from the respective model.

[a]The models control for clustering on hospital- and patient-level and are adjusted for educational attainment, insurance class, type of household, sex, age, chronic condition, comorbidity (NSD and psychic), hospital ward, need of intensive care, discharge destination, language region of hospital and year of discharge

Model D.1 indicated that the observed effect of allophone language was not present in patients hospitalized for colon cancer or COPD, while patients with ischaemic heart disease without command of the regional language stayed 0.35 days (0.096, 0.594, p = 0.007) longer compared to those who spoke the regional language (S2 Table). Regarding migration background, patients of the first generation stayed 0.16 days longer compared to those without any migration background (p = 0.005, Model D.2), although the test of interaction revealed that this effect was not present in patients whose main diagnosis was osteoarthritis. Second generation migrants had equally long stays for most chronic conditions compared to Swiss nationals without a migration background, but they left the hospital earlier if they had a main diagnosis of colon cancer (-1.28 days; -2.17, -0.40; p = 0.005) and stayed longer with the main diagnosis of back problems (0.39 days; 0.09, 0.69, p = 0.011) (Table 5). For all other variables (not related to migration) Models D.1 and D.2 produced similar results to Model D.

In the sensitivity analysis, excluding those patients who had died in hospital (n = 2,198 records), point estimates and the significance of Model D changed only marginally.

The triangle for *educational attainment* and the postulated *mediator comorbidity* (Fig 4) reveals that educational attainment was a significant predictor of inpatients' comorbidity and that both mandatory (a1) and upper secondary education (a2) were associated with higher numbers of side diagnoses compared to tertiary education (S3 Table). On the other hand, as already seen in the Models B-D (Table 4), the number of side diagnoses was significantly associated with longer hospital stays (b). The indirect effect of educational attainment on length of stay is the product of a1*b and a2*b and was significant for both compulsory education (0.334 95% CI: 0.283, 0.388) and upper secondary education (0.206, 0.169, 0.245). Adjusted for number of side diagnoses, the direct effect of educational attainment on length of stay (c1' and c2')

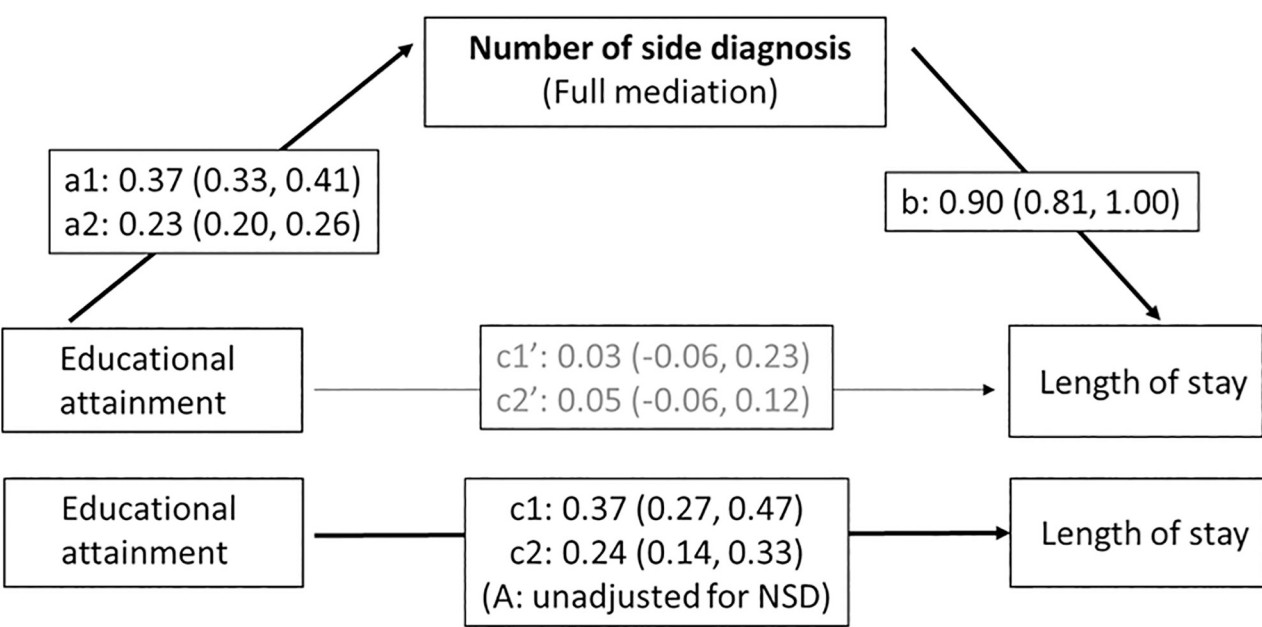

**Fig 4. Mediation of the effect of educational attainment on length of stay by the number of side diagnoses (educational attainment: Compulsory = a1, c1, c1'; upper secondary = a2, c2, c2'; tertiary = reference).** Indirect effects of educational attainment on length of stay: a1*b = 0.371*0.901 = 0.334 (95% Monte Carlo CI: 0.283, 0.388); a2*b = 0.229*0.901 = 0.206 (0.169, 0.245). Mediation Model with intermediate outcome number of side diagnoses: controlling for clustering on hospital- and on patient-level and adjusted for sex, age, nationality, insurance class, household type, chronic condition, language region of hospital and year of discharge.

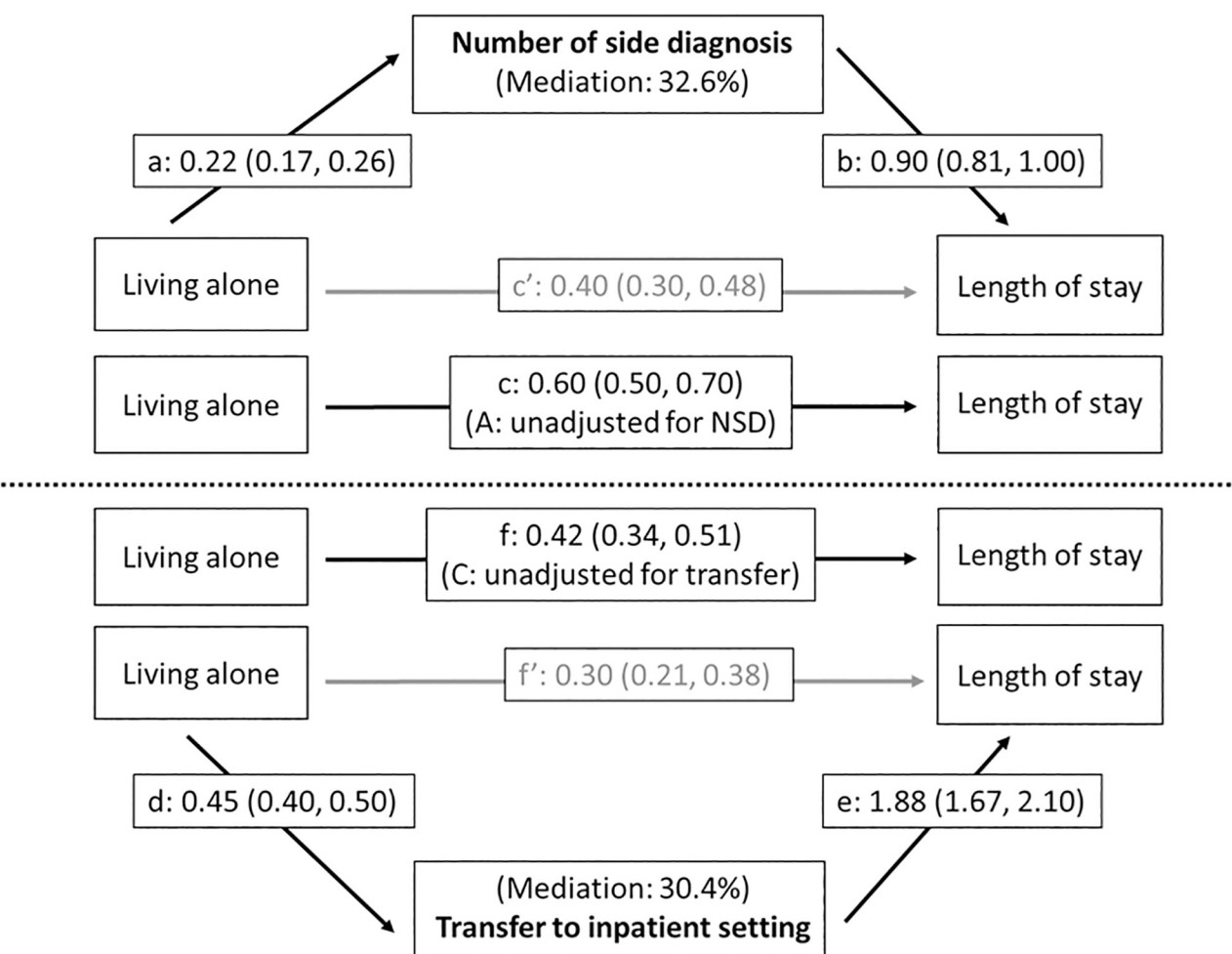

**Fig 5.** Mediation of the effect of living alone on length of stay by the number of side diagnoses (top triangle: a, b, c and c' = coefficients of linear CCMM) and transfer to inpatient setting (bottom triangle: d = coefficient of logistic CCMM; e, f and f' = coefficients of linear CCMM). Indirect effect of living alone on length of stay (via number of side diagnoses): $a*b = 0.216*0.901 = 0.194$ (95% Monte Carlo CI: 0.149, 0.240); indirect effect of living alone on length of stay via transfer to inpatient setting: $zMediation = \frac{z_d z_e}{\hat{\sigma} z_{de}} = 12.62$ (p<0.001). *Mediation Model with intermediate outcome number of side diagnoses (top)*: controlling for clustering on hospital- and on patient-level and adjusted for sex, age, nationality, educational attainment, insurance class, chronic condition, language region of hospital and year of discharge. *Mediation Model with intermediate outcome transfer to inpatient setting (bottom)*: controlling for clustering on hospital- and on patient-level and adjusted for sex, age, nationality, educational attainment, insurance class, chronic condition, number of side diagnoses, psychic comorbidity, hospital ward, need of intensive care, language region of hospital and year of discharge.

was no longer significant. This means that the effect of education on length of stay (c1 and c2) was almost fully mediated by the number of side diagnoses.

Fig 5 illustrates the two postulated mediation pathways regarding social resources. The pathway on top shows that living alone was also a significant predictor of comorbidity (a) (S3 Table) while comorbidity predicted length of stay (b), resulting in a significant indirect effect (0.194, 95%CI: 0.149, 0.240). The direct effect of living alone on length of stay (c) was reduced by 32.6% but remained significant when adjusted for comorbidities (c). The effect of living alone was thus only partially mediated by the burden of comorbidity. The pathway for the postulated *mediator discharge destination* (Fig 5, bottom) shows, that those living alone had a significantly higher risk for transfer to another inpatient setting compared to those living with others (d) (S4 Table), while transfer to another inpatient setting was associated with a longer

hospital stay (e). Following Iacobucci's method [73] for combined linear and logistic regression in mediation analysis, the indirect effect of living alone on length of stay (mediated by discharge destination) was statistically significant (z-test: $z_{Mediation} = 12.62$; p<0.001). Adjustment for discharge destination reduced the effect (f) by 30.4%, which means that the effect of living alone on length of stay was also partially mediated by the transfer to another inpatient setting. Since the adjusted direct effect (f') was still significant, this suggests a third, direct pathway for the impact of living alone on length of stay or the presence of other mediators for which information is missing.

## Main drivers of length of stay

Overall, insurance class, living alone and migration factors were significantly related to the length of hospital stays in the fully adjusted model, but the effect sizes were generally rather small with average increases in length of stay of less than 0.5 days compared to the reference groups. Also, the indirect effects that could be attributed to education and living alone in mediation analysis were within this range. The difference between men and women was in the same order of magnitude, with 0.3 days shorter stays for men, while the oldest age group (75–84 years) stayed 0.7 days longer in hospital compared to the reference age group (25–44 years; S5 Table). Interaction tests indicated that for some chronic conditions and some social groups the differences in length of stay reached up to 1.0 to 1.5 days (S2 Table). In comparison, the effect sizes of medical factors were generally larger. In the fully adjusted model, the main diagnosis and the number of comorbidities were the main drivers of the length of hospital stay: compared to the reference group of patients with ischaemic heart disease (who had the lowest average length of stay of the 15 chronic conditions analysed) those admitted for diabetes with complications stayed 7.5 days longer, on average, and those admitted for colon cancer stayed 6.9 days longer. With each additional side diagnosis, patients stayed on average 0.8 days longer in hospital. Since the test of linearity indicated a linear association between the number of side diagnoses and length of stay (cf. methods section), patients with 13 or more comorbidities stayed on average about 10 days longer in hospital compared to those with no side diagnosis. Lastly staying in the surgical ward as well as the need for intensive care and the transfer to another inpatient setting significantly increased the average hospital stay by about two to three days each.

## Discussion

In this large, representative sample of inpatients hospitalized for chronic conditions in Switzerland, medical factors were the strongest determinants of length of stay in the fully adjusted model (Model D). These included main diagnosis (up to seven days difference), number of comorbidities (up to about 10 days) and treatment-related factors (two to three days). Moreover, we found evidence for differential associations between social factors and length of hospital stays. In general, socially more disadvantaged patients stayed longer in hospital compared to the more privileged, although for insurance class we found some inverse relationships. However, the number of comorbidities also acted as a mediator of the effects of education and living alone on the length of stay while discharge destination was identified as a second mediator of the effect of living alone. When only taking into account demographic factors, main diagnosis and clustering on patient and hospital levels, patients with upper secondary level education stayed 0.24 days (95% CI: 0.14, 0.33) longer and those with compulsory education stayed 0.37 days (95%CI: 0.27, 0.47) longer in hospital compared to those with tertiary education. These effects were almost fully mediated by the burden of comorbidities. The observed

effect of living alone on the length of stay (+0.60 days, 95% CI: 0.50, 0.70) was partially mediated by both the burden of comorbidities (32.6%) and the discharge destination (30.4%).

Only a few studies have published adjusted effect estimates of the impact of social factors on length of single hospital stays in days that may be compared to the present analysis. A study that included a large cohort of acute medical and surgical inpatients reported that the poorest patient group stayed 0.16 days longer in hospital compared to the wealthiest group while Black patients had 0.25 days longer hospital stays than White patients [30]. In a single-hospital study patients with low health literacy were found to stay 0.6 days longer in hospital compared to those with good health literacy [31]. Further, in a study on elderly patients hospitalized for acute care at an internal or geriatric ward, those living alone had 0.72 days longer stays [35]. Despite the methodological differences, these estimates are quite consistent with the effect sizes observed for the social factors in the present study, with adjusted differences between social groups of around 0.1–0.6 days.

The present analysis provides evidence that *educational attainment* per se is a predictor of length of stay in patients hospitalized for some of the investigated chronic conditions. In the fully adjusted model and including an interaction term between education level and main diagnosis, lower educational attainment was associated with longer hospital stays for patients with colon cancer, COPD and asthma and somewhat shorter stays for those with ischaemic heart disease, while for the other chronic conditions no educational gradient was observed. Colon cancer may be detected in a later stage in patients with low education because they make less use of cancer screening such as colonoscopy [74], resulting in longer stays. COPD and asthma are the two classical respiratory ACS conditions for which medication adherence and patient self-management skills are essential for the prevention of hospital admissions [5, 75]. Low health literacy has been found to be associated with both low COPD self-management skills [6, 76] and longer length of stay of COPD-patients [31]. Patients with poor health literacy may therefore need more time to have their diagnosis, treatment and medication explained. Thus, it is plausible that adverse effects of health literacy may also have an impact in the inpatient setting. The SIHOS database, however, does not include an indicator that would measure health-related knowledge or patient self-management abilities rather than general education level. However, the observed inverse effect for ischaemic heart disease patients is less plausible.

For most chronic conditions the effect of educational attainment on the length of stay observed in Model A collapsed when controlling for number of comorbidities. This is in line with the few studies on education and length of stay that found no effect of education level when controlling for demography and for factors related to health status [26, 32]. In this study, however, we could show that the burden of comorbidities almost fully mediated the effect of educational attainment. This implies that patients with compulsory and upper secondary education do have longer hospital stays compared to those with tertiary education, also for other chronic conditions than colon cancer, COPD and asthma. However, their prolonged stays can be attributed to their poorer health status and presumably not to extra time provided for instructions of patients with poor health literacy. Significant associations between education level and cumulative number of bed days observed in studies without adjustment for current health status, treatment in hospital and discharge destination therefore show the total effects of education level without taking into account the different causal pathways [4, 16, 23, 24, 29].

*Hospital insurance class* served as indicator of financial resources in the SIHOS database and thus can be conceived as another indicator of vertical inequality [69]. Interestingly, insurance class was not a predictor of the number of side diagnoses in our analysis (S3 Table), suggesting that, in contrast to the effect of educational attainment, this effect is not mediated by comorbidity (S3 Table). Yet, in the Swiss health system (semi-) private hospital insurance acts

also as financial incentive system and thus may have an impact on the type and amount of diagnostic procedures and treatments in hospital [69, 70]. Overall and in contrast to former studies [28], patients with (semi-)-private insurance were found to have longer stays compared to those with basic insurance in the current analysis, suggesting that they may undergo more diagnostic procedures and/or more treatments. The observed interaction between insurance class and main diagnosis—with longer stays among semi-privately insured patients with AMI, COPD and back problems and shorter stays among respective patients with colon and breast cancer—suggests two different mechanisms: (1) for certain chronic conditions (semi-)-private insurance may facilitate access to specific medical procedures that are related to additional hospital days. (2) more financial resources and (semi-)-private insurance may imply better access to preventive measures such as mammography and colonoscopy resulting in earlier detection of breast and colon cancer, allowing less invasive treatment and earlier discharge [74, 77, 78]. However, to better disentangle effects related to insurance class and to differentiate between social gradients, effects of financial incentives and preventive screening behaviour, more in-depth analysis of specific conditions and particular treatments, which go beyond the scope of the current analysis, would be necessary.

*Living alone* as opposed to living with others has been discussed as proxy for different aspects of social and health-related resources [79]. It has been associated with negative aspects such as limited social support, worse health status, poor adherence and non-use of medications in COPD-patients [80] and higher risk of morbidity and mortality [18, 19]. A positive aspect is the better functional status of older persons living on their own [35, 37]. Thus, the association between the indicator living alone and length of hospital stay may be complex. The present analysis could disentangle three pathways of the effect of social resources on length of stay, each explaining about one third of the total effect. The first indirect path is mediated by the burden of comorbidity and additional hospital days can most probably be attributed to the poorer health status of those living alone. Better functional status in the elderly, allowing an independent live [35, 37], may therefore not necessarily imply fewer comorbidities. The second indirect path is mediated by transfer to another inpatient setting, with additional hospital days probably explained by the time needed to seek a suitable place or waiting time until a place is available in an appropriate institution. The third and last path suggests a direct effect of living alone that may be explained by extra time needed until the patient is sufficiently independent to cope at home alone, although the presence of further mediators for which information is missing cannot be excluded.

The three indicators related to a *migration background* of patients, namely nationality, migration status and language skills, were all associated with longer hospital stays in the fully adjusted model. Significant effects were observed for allophone patients, first generation migrants and non-EU/EFTA nationals, i.e., those migration groups with probably the poorest integration and most pronounced cultural differences to the host country. The strongest effects were observed for the allophone patients, i.e., for those with the poorest language skills, and for non-EU/ EFTA nationals, while for those who speak at least one official language or English only ischaemic heart disease patients had prolonged stays. For second or higher order generation migrants there was no evidence for prolonged hospital stays.

The time needed to organize interpreter services may at least partially explain prolonged stays of allophone patients, but also of first-generation migrants and non-EU/EFTA nationals, given the collinearities between the three migration factors (cf. statistical analysis). However, the use of interpreter services has been found to be associated with both shorter [43, 44] and longer [44] hospital stays. The generally poorer health status of patients with limited language skills [11] or the selective use of interpreters for medically more complex patients [45, 46] may

also contribute to longer hospital stays of allophone patients. In the current analysis this is less likely, having controlled for the burden of comorbidities and the use of intensive care.

The implementation of community interpreting in Switzerland (https://www.inter-pret.ch) is still considered insufficient [81] since only some hospitals in Switzerland routinely rely on community interpreters instead of ad-hoc interpreters [38]. Community interpreting that goes beyond classic word-for-word interpreting and includes intercultural explanations, building patient-provider relationships and accompanying immigrant patients [82], is expected to improve both access to the health care system and adequate use of health care services of patients with language and cultural barriers [81]. However, the impact of availability of interpreting services in a hospital on length of hospital stays of poorly integrated patients could not be analysed in this study, since the SIHOS database does not include the pertinent information.

In summary, this analysis of a large and representative sample of inpatients hospitalized for chronic conditions in Switzerland identified health-related aspects as well as factors on the individual, organizational and system level that may explain the impact of social factors on length of hospital stays. Health-related aspects turned out to be the main drivers of length of stay. This includes primarily the main diagnosis and the burden of comorbidity, the latter with additional mediation effects, but also treatments related to the health problem. On the individual level, the availability of support at home and the degree of independence seem to be taken into account for discharge decisions and may explain the direct effect of living alone on the length of stay identified in mediation analysis. On the organizational level, time needed to organize transfers or interpreter services may explain prolonged hospital stays for patients not discharged to their homes and for poorly integrated migrants, respectively. Finally, on the system level, financial incentives, insufficient implementation of community interpreter services or lacking support for patients with low health literacy are factors that may lead to over- or underuse of diagnostic procedures and treatments both before and during the hospital stay. For example, patients who forego preventive measures such as colonoscopy due to poor health literacy or for financial reasons may start treatment in a later cancer stage, as suggested by the prolonged hospital stays for colon cancer in patients with low educational level and basic insurance. The prolonged stays for COPD and asthma patients with low education may reflect extra time needed for additional support but could also be related to a more severe stage of disease not captured with the number of side diagnoses [83]. The current analysis does not provide direct evidence for premature discharge of socially disadvantaged patients. Nevertheless, equally long stays of patients with low education level compared to well-educated patients with generally better health literacy skills [84] suggest that inpatients with poor health literacy may not always receive adequate additional support, which would imply somewhat longer hospital stays.

## Strengths and limitations

One of the unique strengths of the present analysis is that it was based on a large and representative sample of inpatients hospitalized for acute care for highly relevant chronic conditions and that it could rely on individual-level information about medical, demographic and social parameters. Unlike some of the previous studies investigating the impact of social determinants on length of hospital stays, our analysis was neither based on aggregated data of social groups [16, 23, 29, 30] nor on cumulative bed-days [4, 16, 23, 29]. Further, it was neither restricted to one hospital [31, 32] nor to a single chronic condition [16, 26].

The limitations of our analysis are mainly related to the implications of administrative data that are not tailored to the study question. Therefore, some unmeasured confounding may be

present. A first possible source of bias could be the period of time between participation in the Structural Survey and the hospital stay. This timespan may have reached a maximum difference of five years (e.g., participation in SE 2010 and hospitalisation in December 2016) that may have led to misclassification bias for the variables derived from the Structural Survey. While migration status and educational attainment [67] are expected to be stable over time, the type of household and language skills may have changed for some patients between participation in the SE and hospitalisation. Such misclassification would result in bias towards the null [85] and our study would underestimate the effect of living alone and of allophone language skills on length of stay. Further, a meta-analytical review of the influence of social relationships on mortality established that simple single-item measures such as living alone versus not living alone seem to underestimate associations with social resources [19]. Possible bias due to this simple indicator would therefore go in the same direction as possible bias due to asynchronous assessment. The concerns regarding asynchrony do not apply to insurance class or the demographic variables of age, sex, and nationality, since these variables were assessed during the hospital stay.

Another possible source of bias concerns the hospital level. No information was available regarding support of disadvantaged patients, organization of discharge or translation services. Although all CCMM controlled for clustering on the hospital level, unmeasured confounding due to differences between hospitals regarding these variables cannot fully be ruled out.

The CCMM controlled for differential health status of inpatients with the three variables main diagnosis, number of comorbidities and psychic comorbidity. However, for lung cancer there is evidence that comorbidity is not associated with the stage of cancer [86]. Therefore, adjustment only for comorbidity may not fully control for the impact of health status on length of stay in cancer patients. However, information on stage of cancer is not available in the SIHOS database.

There is evidence from Germany that the introduction of the DRG-system has led to changes in the coding of comorbidity [87]. Because in our data an increasing number of side diagnoses was also observed over time in the MS, the year of discharge was included in all CCMM but may not have fully adjusted for this change of system or for secular trends.

Although some problems with erroneous anonymous linkage codes were identified during validation of the database, there is evidence that the 30 percent mismatches should not seriously affect the analysis of social gradients and the comparison of different groups in the SIHOS inpatient cohort, since the mismatches were randomly distributed with regard to most variables of interest [58]. The observed underrepresentation of non-European migration groups most probably can be explained with misspelling of unfamiliar names resulting in mismatches due to the hashing procedure [56]. The reported effect estimates for migration factors, however, would only be biased if they differed between patient groups with more or less complicated names, which is rather unlikely.

## Conclusions

We conclude that inpatient care in Switzerland seems to take rather obvious individual needs of patients into account, such as extra time for those living alone or to organize a transfer or an interpreter, but not necessarily more hidden needs of patients with low health literacy and fewer resources to assert their interests within the health system. However, hospital admission could open a window of opportunity to discern these patients and to provide them with extra time and support to improve their self-management skills and to better cope with everyday life after discharge, thus reducing the risk of future hospital stays particularly related to ACS-conditions [3]. Further, on the level of the health care system, financial incentives and access

barriers seem to result in prolonged hospital stays for some patients that put a financial burden on the health system and, in the worst case, result in inadequate treatments of patients and adverse health outcomes. These findings underpin the importance attributed to health policies promoting shared decision making and patient-centred care [88]. They should prompt the development and implementation of measures tailored to the differential needs of social and cultural groups, including:

- Screening of social situation on hospital admission [89] to identify patients who need social support or more instructions and to provide them with extra time needed

- Interprofessional discharge planning to identify patients who will need a transfer or support at home and to organize their timely discharge, reducing unnecessary and costly acute hospital days [90, 91]

- Access to community interpreting in all hospitals for patients with lack of local language skills at admission, during hospital stay and at discharge to assure adequate use of health care structures [81]

- Facilitation of access to cancer prevention measures, e.g., with mammography and colonoscopy that are paid by basic insurance without franchise or out-of-pocket payments, and with screening programs that are tailored for socially disadvantaged and migration groups.

Future research should address open questions e.g., related to the longer hospital stays of (semi)-privately insured patients with a main diagnosis of AMI, COPD and back problems and disentangle the impact of social factors, insurance class and financial incentives on treatments and length of stay.

## Supporting information

**S1 Fig.** Length of stay by age and social factors (top left: sex; top right: education; middle left: insurance class, middle right: type of household; bottom right: language skills; bottom left: migration background.
(TIF)

**S2 Fig.** Comorbidity by age and social factors (top left: sex; top right: education; middle left: insurance class, middle right: type of household; bottom right: language skills; bottom left: migration background).
(TIF)

**S3 Fig.** Discharge destination by age and social factors (top left: sex; top right: education; middle left: insurance class, middle right: type of household; bottom right: language skills; bottom left: migration background).
(TIF)

**S1 Table. Cluster sizes and Intra-Class-Correlation (ICC) of cluster variables, linear CCMM (null-model with outcome length of stay).**
(PDF)

**S2 Table. Effect estimates for significant interactions between main diagnosis and social factors.** The least significant difference adjusted significance level is 0.05. [1]) cf. Table 4; [2]) cf. Table 5.
(PDF)

**S3 Table. Associations of comorbidity with social factors (linear CCMM).** β: Difference in average number of comorbidities to the respective reference category estimated from the

mixed linear regression model containing all variables listed in the table and random effects for hospitals and patients. +centred by chronic conditions; *Controlling for clustering on hospital- and patient-level and adjusted for age, chronic condition, language region of hospital and year of discharge.
(PDF)

**S4 Table. Associations of discharge destination with social factors and factors related to hospital stay (logistic CCMM).** *Controlling for clustering on hospital- and patient-level and adjusted for age, chronic condition, language region of hospital and year of discharge.
(PDF)

**S5 Table. Association of length of stay with demographic factors (age implemented as categorical variable), social factors, health status and factors related to hospital stay (linear CCMM D).** The regression coefficients β are the estimated differences in average length of stay between the respective category and the reference category obtained from the respective model. *The model controls for clustering on hospital- and on patient-level and is adjusted for language region of hospital and year of discharge.
(PDF)

**S1 File. Description of data sources and specification of variables.**
(PDF)

## Acknowledgments

Members of the SIHOS Team: Lucy Bayer-Oglesby[1]* (PI), Nicole Bachmann[1], Andrea Zumbrunn[1], Maria Solèr[1], Marcel Widmer[2], Reto Jörg[2], Carlos Quinto[3], Christian Schindler[3], Daniel Zahnd[4]

[1]Institute for Social Work and Health, FHNW School of Social Work, Olten, Switzerland; [2]Swiss Health Observatory, Neuchâtel, Switzerland; [3]Swiss Tropical and Public Health Institute, Basel, Switzerland; [4]InfoNavigation, Bern, Switzerland

*Head of the SIHOS-Team

E-mail: lucy.bayer@fhnw.ch (LBO)

## Author Contributions

**Conceptualization:** Lucy Bayer-Oglesby, Andrea Zumbrunn, Nicole Bachmann.

**Data curation:** Lucy Bayer-Oglesby, Andrea Zumbrunn, Nicole Bachmann.

**Formal analysis:** Lucy Bayer-Oglesby, Andrea Zumbrunn, Nicole Bachmann.

**Funding acquisition:** Lucy Bayer-Oglesby, Andrea Zumbrunn, Nicole Bachmann.

**Methodology:** Lucy Bayer-Oglesby, Andrea Zumbrunn, Nicole Bachmann.

**Project administration:** Lucy Bayer-Oglesby.

**Validation:** Lucy Bayer-Oglesby, Andrea Zumbrunn, Nicole Bachmann.

**Visualization:** Lucy Bayer-Oglesby.

**Writing – original draft:** Lucy Bayer-Oglesby.

**Writing – review & editing:** Lucy Bayer-Oglesby, Andrea Zumbrunn, Nicole Bachmann.

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
