## [Decision Letter · Decision Letter 0]

9 Mar 2022

PONE-D-21-31664Social inequalities, length of hospital stay for chronic conditions and the mediating role of comorbidity and discharge destination: A multilevel analysis of hospital administrative data linked to the population census in SwitzerlandPLOS ONE

Dear Dr. Bayer-Oglesby,

Thank you for submitting your manuscript to PLOS ONE. After careful consideration, we feel that it has merit but does not fully meet PLOS ONE’s publication criteria as it currently stands. Therefore, we invite you to submit a revised version of the manuscript that addresses the points raised during the review process.

We look forward to receiving your revised manuscript.

Kind regards,

Dong Keon Yon, MD, FACAAI

Academic Editor

PLOS ONE

Journal Requirements:

3. One of the noted authors is a group or consortium SIHOS Team. In addition to naming the author group, please list the individual authors and affiliations within this group in the acknowledgments section of your manuscript. Please also indicate clearly a lead author for this group along with a contact email address.

Additional Editor Comments:

Thank you for submitting your manuscript to Plos One. The reviewers and I believe it is of potential value for our readers. However, the reviewers have raised a number of very important issues, and their excellent comments will need to be adequately addressed in a revision before the acceptability of your manuscript for publication in the Journal can be determined. We cannot guarantee that your revised paper will be chosen for publication; this would be solely based on how satisfactorily you have addressed the reviewer comments.

Reviewers' comments:

Reviewer's Responses to Questions

**Comments to the Author**

1. Is the manuscript technically sound, and do the data support the conclusions?

Reviewer #1: Yes

Reviewer #2: Partly

Reviewer #3: No

2. Has the statistical analysis been performed appropriately and rigorously? 

Reviewer #1: Yes

Reviewer #2: Yes

Reviewer #3: No

3. Have the authors made all data underlying the findings in their manuscript fully available?

Reviewer #1: Yes

Reviewer #2: Yes

Reviewer #3: Yes

4. Is the manuscript presented in an intelligible fashion and written in standard English?

Reviewer #1: Yes

Reviewer #2: Yes

Reviewer #3: No

5. Review Comments to the Author

Reviewer #1: The abstract is concise and highly informative and contains all the elements that should be found in an ideal abstract.

The introduction clearly states the questions needed to be addressed in the study and sets the stage for the rest of the study gradually and by including relevant literature.

The variables were explicitly stated and defined clearly.

It also clearly defined that the unit of analysis was the individual rather than aggregates or groups of individual which is in itself a strength of the study and what makes it unique from other studies in the literature.

The source and sampling strategy were also clearly stated.

Statistical methods were described in a very detailed manner including controlling for confounding and subgroup interactions. Missing data were also addressed.

All in all, it was a very detailed well written paper, to which the non expert can understand its contents.

Reviewer #2: This an important study for the patients' wellbeing.

In Abstract please mention the analysis procedure.

STROBE and RECORD abbreviations need explanation.

Give some precise outcome based recommendations and suggest some way outs.

Reviewer #3: First, your interest was very impressive but you lost the Plose one manuscript writing format and your methodology is poorly written

your mode of reporting like figure and table does not match with the document you report.

To large sentence at one paragraph and font size and format lacks standard

The whole manuscript is full of grammatical errors and pleases re-write and please see Plose one manuscript writing format and policies before submitting it.

in sum, your manuscript lacks scientific manuscript writing format and you upload what you write in your research paper and it makes the reader bored.

INTRODUCTION: too vast lacks clarity and is not separated into paragraphs and lacks the gap & reason/justification of the study.

methods: poorly written and lacks the source population and study population. it is not also clear how study participants were selected. and it is difficult to say .research because it lacks sampling.

result: it is not well written like response rate, outcome variable, and independent variables were not clearly reported.

There is a list of figures in the text but no figure.

Discussion: I don't know which type of discussion writing style/format you used. It lacks comparison with other studies, justification for significant variables, and implication of the factors for policymakers.

limitation and strength of the study: not stated

conclusion: seems discussion

generally, please try to other works

Associations Between Social Factor Documentation and Hospital Length of Stay and Readmission Among Children https://pubmed.ncbi.nlm.nih.gov/31888952/

6. PLOS authors have the option to publish the peer review history of their article (what does this mean?). If published, this will include your full peer review and any attached files.

Reviewer #1: No

Reviewer #2: **Yes: **Dr. Nasrin Akter

Reviewer #3: No

---

## [Author Response · Author response to Decision Letter 0]

26 Apr 2022

PONE-D-21-31664

Social inequalities, length of hospital stay for chronic conditions and the mediating role of comorbidity and discharge destination: A multilevel analysis of hospital administrative data linked to the population census in Switzerland

Journal Requirements:

Response: We ensured that our manuscript meets PLOS ONE’s style requirements and file naming guidelines. 

Response: The data underlying our results are owned by a third party and we do not have the permission to share the data. We have adapted the data availability statement to conform with the sample text for third-party data, including the necessary contact information that others will need to access the data in the same manner as we did (see Cover Letter). Further, we upload a supporting information file (S1_File) containing the description of the data sources and the specification of the variables that need to be requested from the third-party.

3. One of the noted authors is a group or consortium SIHOS Team. In addition to naming the author group, please list the individual authors and affiliations within this group in the acknowledgments section of your manuscript. Please also indicate clearly a lead author for this group along with a contact email address.

Response: We have added this information 

Response: We have moved the list of supporting tables and figures from the supporting-information file to the end of the manuscript and completed the list of the captions. We have uploaded separate files for the supplementary tables (S1_Table, S2_Table, S3_Table, S4_Table and S5_Table) and a new S1_File with the supporting information regarding data availability (see 2.). 

Additional Editor Comments:

Thank you for submitting your manuscript to Plos One. The reviewers and I believe it is of potential value for our readers. However, the reviewers have raised a number of very important issues, and their excellent comments will need to be adequately addressed in a revision before the acceptability of your manuscript for publication in the Journal can be determined. We cannot guarantee that your revised paper will be chosen for publication; this would be solely based on how satisfactorily you have addressed the reviewer comments.

Response: Thank you very much for your time and careful review of our manuscript. The comments helped us to improve the paper. 

Reviewers' comments:

Reviewer's Responses to Questions

Comments to the Author

1. Is the manuscript technically sound, and do the data support the conclusions?

Reviewer #1: Yes

Reviewer #2: Partly

Reviewer #3: No

2. Has the statistical analysis been performed appropriately and rigorously? 

Reviewer #1: Yes

Reviewer #2: Yes

Reviewer #3: No

3. Have the authors made all data underlying the findings in their manuscript fully available?

Reviewer #1: Yes

Reviewer #2: Yes

Reviewer #3: Yes

4. Is the manuscript presented in an intelligible fashion and written in standard English?

Reviewer #1: Yes

Reviewer #2: Yes

Reviewer #3: No

5. Review Comments to the Author

Reviewer #1: The abstract is concise and highly informative and contains all the elements that should be found in an ideal abstract.

Response: Thank you for this observation 

The introduction clearly states the questions needed to be addressed in the study and sets the stage for the rest of the study gradually and by including relevant literature.

Response: Thank you for the appreciation of our argumentation line 

The variables were explicitly stated and defined clearly.

Response: Thank you

It also clearly defined that the unit of analysis was the individual rather than aggregates or groups of individual which is in itself a strength of the study and what makes it unique from other studies in the literature.

Response: Thank you, this appreciates our great efforts to create this unique database for Switzerland

The source and sampling strategy were also clearly stated.

Response: Thank you

Statistical methods were described in a very detailed manner including controlling for confounding and subgroup interactions. Missing data were also addressed.

Response: Thank you

All in all, it was a very detailed well written paper, to which the non expert can understand its contents.

Response: We are very thankful for your comments on our manuscript and your supportive feedback. 

Reviewer #2: This an important study for the patients' wellbeing.

Response: We are very thankful for your careful review of our manuscript and valuable comments, which helped to improve it. 

In Abstract please mention the analysis procedure.

Response: In the abstract we mention that we performed cross-classified multilevel models. Now we additionally mention the mediation analysis (unmarked version lines 22-23/marked-up copy lines 22-23). However, we did not add further details on the analysis procedure in the abstract to comply with the submission guidelines of PLOS ONE (no methodological detail in the abstract, max. length 300 words). 

STROBE and RECORD abbreviations need explanation.

Response: We have written out the full terms now (lines 158-160/210-212)

Give some precise outcome based recommendations and suggest some way outs.

Response: Thank you for this inspiring remark, we have extended the conclusion section as follows (lines 725 ff./791 ff.): 

…These findings underpin the importance attributed to health policies promoting shared decision making and patient-centred care [85]. They should prompt the development and implementation of measures tailored to the differential needs of social and cultural groups, including:

• Screening of social situation on hospital admission [88] to identify patients who need social support or more instructions and to provide them with extra time needed 

• Interprofessional discharge planning to identify patients who will need a transfer or support at home and to organize their timely discharge, reducing unnecessary and costly acute hospital days [86,87]

• Access to community interpreting in all hospitals for patients with lack of local language skills at admission, during hospital stay and at discharge to assure adequate use of health care structures [78]

• Facilitation of access to cancer prevention measures, e.g., with mammography and colonoscopy that are paid by basic insurance without franchise or out-of-pocket payments, and with screening programs that are tailored for socially disadvantaged and migration groups. 

Future research should address open questions e.g., related to the longer hospital stays of (semi)-privately insured patients with a main diagnosis of AMI, COPD and back problems and disentangle the impact of social factors, insurance class and financial incentives on treatments and length of stay.

New References:

86. Koch D, Schuetz P, Haubitz S, Kutz A, Mueller B, Weber H, et al. Improving the post-acute care discharge score (PACD) by adding patients’ self-care abilities: A prospective cohort study. PLOS ONE. 2019 Mar 28;14(3):e0214194. 

87. Kutz A, Koch D, Conca A, Baechli C, Haubitz S, Regez K, et al. Integrative hospital treatment in older patients to benchmark and improve outcome and length of stay – the In-HospiTOOL study. BMC Health Services Research. 2019 Apr 23;19(1):237. 

88. Pantell MS, Kaiser SV, Torres JM, Gottlieb LM, Adler NE. Associations Between Social Factor Documentation and Hospital Length of Stay and Readmission Among Children. Hospital Pedi-atrics. 2020 Jan 1;10(1):12–9.

 

Reviewer #3: First, your interest was very impressive but you lost the Plose one manuscript writing format and your methodology is poorly written

Response: We appreciate your remarks that helped to improve the method section (amendments are described below, following your more detailed comments on the methods section). 

your mode of reporting like figure and table does not match with the document you report.

Response: We have discussed this in our group, and it remained unclear to us where you saw a mismatch. The figures and tables are explained in the manuscript: Fig 1 is referred to in line 180/233 and is used to illustrate the data sources and the data linkage of the SIHOS database as well as the selection process of the study sample (line 176 ff./229 ff.). Fig 2 illustrates the cross-clustering nature of our data and is referred to in line 305/363 of our manuscript. In the results section we introduce Fig 3 (line 362 ff./424 ff.), Fig 4 (line 454 ff./516 ff.) and Fig 5 (line 473 ff./535 ff.). 

To large sentence at one paragraph and font size and format lacks standard

Response: Thank you for these observations. We have checked the paper for long sentences and made amendments. We revised font size and formats and double-checked that they meet PLOS ONE’s style requirements. 

The whole manuscript is full of grammatical errors and pleases re-write and please see Plose one manuscript writing format and policies before submitting it.

Response: We double-checked the manuscript to comply with Standard English (British English) and with PLOS ONE writing format and policies. 

in sum, your manuscript lacks scientific manuscript writing format and you upload what you write in your research paper and it makes the reader bored.

Response: Thank you for these general comments. We suppose you refer to our previous publication “Social situation and hospitalisation due to chronic conditions” [3], where we made use of the same database as the submitted manuscript. In the introduction we took up some general arguments on social inequalities and health that we shortened now (see our response to your next comment). Besides this, the current manuscript is on a different topic (length of hospital stay instead of hospitalization risks), uses a different subsample of the SIHOS-database (patient sample instead of population sample) and has a different analytic approach (cross-classified multilevel models and mediation analysis vs. multivariate logistic regression). 

3. Bayer-Oglesby L, Bachmann N, Zumbrunn A. Social situation and hospitalisation due to chronic conditions | OBSAN [Internet]. 2020 [cited 2021 Mar 9]. Available from: https://www.obsan.admin.ch/en/publications/social-situation-and-hospitalisation-due-chronic-conditions

INTRODUCTION: too vast lacks clarity and is not separated into paragraphs and lacks the gap & reason/justification of the study.

Response: Thank you for these constructive remarks. We have restructured the introduction, moved some parts to the discussion, separated it into more paragraphs and shortened it. We put now more emphasis on the research gap and justification of the study (lines 119 ff./166 ff.). 

methods: poorly written and lacks the source population and study population. it is not also clear how study participants were selected. and it is difficult to say .research because it lacks sampling.

result: it is not well written like response rate, outcome variable, and independent variables were not clearly reported.

Response: Thank you for these remarks. The source population is described in the method section, line 168/220 (Swiss population aged 15 and over). The study population of the SIHOS database is defined in the method section, lines 183-185/236-239, lines 195-196/249-250 and Fig 1. It consists of the participants of the structural survey 2010-2014. The selection of the study sample is described in lines 196-199/250-253. To clarify the sampling of the structural survey, performed by the Federal Statistical Office, we have added the response rate (line 167/219), referring to a report, where the sampling of the structural survey is described in detail [53]. 

53. Potterat J, Qualité L, Assoulin D. Strukturerhebung der eidgenössischen Volkszählung: Stichprobenplan und -ziehung, Gewichtung, Schätzverfahren und Pooling 2010-2018 [Internet]. Neuchâtel; 2019. Available from: https://www.bfs.admin.ch/bfs/de/home/statistiken/kataloge-datenbanken/publikationen.assetdetail.11187024.html

The outcome variable is defined under the level 3 heading “Outcome” (line 220/276): “Length of hospital stay (LOS) was based on SwissDRG definition, calculated by day of admission and each subsequent day without the day of discharge and excluding days of leave”. We have realized that the first paragraph in the section “Statistical analysis” may have been confusing regarding the outcome variable (lines 301-302/359-360). We have clarified that we report descriptive statistics of the outcome variable “length of stay” and of the two mediators “number of side diagnoses” and “transfer to inpatient setting”. Accordingly, we report now in table 2 and table 3 the percentage of transfers to inpatient settings instead of the percentage of home discharges, since transfer to inpatient setting was used as intermediate outcome in mediation analysis. 

There is a list of figures in the text but no figure.

Response: Thank you for this comment. Following the PLOS ONE Manuscript Body Formatting Guidelines we are listing the figure captions directly after the paragraph in which they are first cited. The figures are displayed at the end of the manuscript-pdf, as the figure files had to be uploaded separately as individual files during the submission process. 

Discussion: I don't know which type of discussion writing style/format you used. It lacks comparison with other studies, justification for significant variables, and implication of the factors for policymakers.

Response: Thank you for your considerations. In the first paragraph of the discussion, we present a summary of the main findings, as is common in scientific publications. Unfortunately, none of the former studies on length of hospital stay and social factors is directly comparable to the present analysis because of methodological differences. To give at least an idea whether the effect sizes observed in the present study are within the range of former observations, we added a paragraph that compares our results with the few studies that reported the same unit (differences in number of days of a single hospital stay):

“Only a few studies have published adjusted effect estimates of the impact of social factors on length of single hospital stays in days that may be compared to the present analysis. A study that included a large cohort of acute medical and surgical inpatients reported that the poorest patient group stayed 0.16 days longer in hospital compared to the wealthiest group while Black patients had 0.25 days longer hospital stays than White patients [30]. In a single-hospital study, patients with low health literacy were found to stay 0.6 days longer in hospital compared to those with good health literacy [31]. Further, in a study on elderly patients hospitalized for acute care at an internal or geriatric ward, those living alone had 0.72 days longer stays [35]. Despite the methodological differences, these estimates are quite consistent with the effect sizes observed for social factors in the present study, with adjusted differences between social groups of around 0.1-0.6 days.”

30. Ghosh AK, Geisler BP, Ibrahim S. Racial/ethnic and socioeconomic variations in hospital length of stay. Medicine (Baltimore). 2021 May 21;100(20):e25976. 

31. Jaffee EG, Arora VM, Matthiesen MI, Meltzer DO, Press VG. Health Literacy and Hospital Length of Stay: An Inpatient Cohort Study. Journal of Hospital Medicine [Internet]. 2017 Dec 1 [cited 2021 Jun 15];12(12). Available from: https://www.journalofhospitalmedicine.com/jhospmed/article/152314/hospital-medicine/health-literacy-and-hospital-length-stay-inpatient-cohort

35. Agosti P, Tettamanti M, Vella FS, Suppressa P, Pasina L, Franchi C, et al. Living alone as an independent predictor of prolonged length of hospital stay and non-home discharge in older patients. European Journal of Internal Medicine. 2018 Nov 1;57:25–31. 

The following four paragraphs of the discussion focus on one social characteristic each: (1) educational attainment, (2) hospital insurance, (3) living alone and (4) migration background. For each factor we discuss possible explanations, that we underpin with findings of other studies. The next paragraph changes the perspective and first discusses health related aspects and then factors at the individual, organizational and system level that may explain the impact of social factors on length of stay, referring to former studies. Regarding the implication of the factors for policymakers, we have extended the conclusions with recommendations derived from our findings (see our response to reviewer #2). 

limitation and strength of the study: not stated

Response: Limitations and strengths of the study are mentioned and discussed under the level 2 heading “Strengths and limitations” (lines 667 ff./732 ff.). 

conclusion: seems discussion

Response: As already mentioned above, we have extended the conclusions with recommendations 

generally, please try to other works

Associations Between Social Factor Documentation and Hospital Length of Stay and Readmission Among Children https://pubmed.ncbi.nlm.nih.gov/31888952/

Response: Thank you for this valuable suggestion. The paper was not referred to because it focusses on paediatric patients and on social factors related to the family situation of the children that are assessed with the ICD social risk code documentation. We recognize that the discussion regarding documentation of social factors can be extended to adult inpatients, and we have addressed this in our recommendations at the end of the Conclusion section, where we refer to the above-mentioned publication.

6. PLOS authors have the option to publish the peer review history of their article (what does this mean?). If published, this will include your full peer review and any attached files.

Do you want your identity to be public for this peer review? For information about this choice, including consent withdrawal, please see our Privacy Policy.

Reviewer #1: No

Reviewer #2: Yes: Dr. Nasrin Akter

Reviewer #3: No

---

## [Decision Letter · Decision Letter 1]

12 Jun 2022

PONE-D-21-31664R1Social inequalities, length of hospital stay for chronic conditions and the mediating role of comorbidity and discharge destination: A multilevel analysis of hospital administrative data linked to the population census in SwitzerlandPLOS ONE

Dear Dr. Bayer-Oglesby,

Thank you for submitting your manuscript to PLOS ONE. After careful consideration, we feel that it has merit but does not fully meet PLOS ONE’s publication criteria as it currently stands. Therefore, we invite you to submit a revised version of the manuscript that addresses the points raised during the review process.

We look forward to receiving your revised manuscript.

Kind regards,

Dong Keon Yon, MD, FACAAI

Academic Editor

PLOS ONE

Journal Requirements:

Additional Editor Comments (if provided):

The Reviewers and I noted several issues not least of which was methodological concerns. I think that statistical reference may be needed (i.e., https://doi.org/10.54724/lc.2022.e3). Thank you.

Reviewers' comments:

Reviewer's Responses to Questions

**Comments to the Author**

1. If the authors have adequately addressed your comments raised in a previous round of review and you feel that this manuscript is now acceptable for publication, you may indicate that here to bypass the “Comments to the Author” section, enter your conflict of interest statement in the “Confidential to Editor” section, and submit your "Accept" recommendation.

Reviewer #3: All comments have been addressed

2. Is the manuscript technically sound, and do the data support the conclusions?

Reviewer #3: Partly

3. Has the statistical analysis been performed appropriately and rigorously? 

Reviewer #3: No

4. Have the authors made all data underlying the findings in their manuscript fully available?

Reviewer #3: No

5. Is the manuscript presented in an intelligible fashion and written in standard English?

Reviewer #3: No

6. Review Comments to the Author

Reviewer #3: Comments were partially addressed but the document preparation is not following PLoS one criteria. figures and tables were not correctly cited and not well prepared. it needs revision and grammar errors are serious problems.

the flow of ideas, neatness, result reporting form, and discussion are not as such good. statistical analysis is not clearly stated and does not answer the objective.

generally, the way that the manuscript is prepared is difficult to understand.

7. PLOS authors have the option to publish the peer review history of their article (what does this mean?). If published, this will include your full peer review and any attached files.

Reviewer #3: No

---

## [Author Response · Author response to Decision Letter 1]

27 Jun 2022

PONE-D-21-31664

Social inequalities, length of hospital stay for chronic conditions and the mediating role of comorbidity and discharge destination: A multilevel analysis of hospital administrative data linked to the population census in Switzerland

Journal Requirements:

Changes to the reference list

Added references

47. Bachmann N, Zumbrunn A, Bayer-Oglesby L. Social and Regional Factors Predict the Likelihood of Admission to a Nursing Home After Acute Hospital Stay in Older People With Chronic Health Conditions: A Multilevel Analysis Using Routinely Collected Hospital and Census Data in Switzerland. Frontiers in Public Health. 2022;10:871778 

48. Zumbrunn A, Bachmann N, Bayer-Oglesby L, Joerg R. Social disparities in unplanned 30-day readmission rates after hospital discharge in patients with chronic health conditions: A retrospective cohort study using patient level hospital administrative data linked to the population census in Switzerland [Internet]. medRxiv; 2022 [cited 2022 Jan 25]. Available from: https://www.medrxiv.org/content/10.1101/2022.01.18.22269480v1

71. Lee SW. Regression analysis for continuous independent variables in medical research: statistical standard and guideline of Life Cycle Committee. Life Cycle [Internet]. 2022 [cited 2022 Jun 20];2. Available from: http://www.elifecycle.org/archive/view_article?pid=lc-2-0-3

Corrections (as necessary): 

All Journal name abbreviations are now those found in the National Center for Biotechnology Information (NCBI) databases 

All published articles are now cited with traditional volume and page numbers 

Additional Editor Comments (if provided):

The Reviewers and I noted several issues not least of which was methodological concerns. I think that statistical reference may be needed (i.e., https://doi.org/10.54724/lc.2022.e3). Thank you.

Response: We have added the suggested reference at line 312. At line 164 we refer to the paper of Barker et al (2020), which includes recommendations for best practice of cross-classified multilevel analysis. In addition, we now cite Barker also at line 309. At lines 344 and 350 we refer to Hayes and Todd (2018) “Introduction to Mediation, Moderation, and Conditional Process Analysis” 

Reviewers' comments:

Reviewer's Responses to Questions

Comments to the Author

1. If the authors have adequately addressed your comments raised in a previous round of review and you feel that this manuscript is now acceptable for publication, you may indicate that here to bypass the “Comments to the Author” section, enter your conflict of interest statement in the “Confidential to Editor” section, and submit your "Accept" recommendation.

Reviewer #3: All comments have been addressed

2. Is the manuscript technically sound, and do the data support the conclusions?

Reviewer #3: Partly

3. Has the statistical analysis been performed appropriately and rigorously? 

Reviewer #3: No

4. Have the authors made all data underlying the findings in their manuscript fully available?

Reviewer #3: No

Response: We had updated the Data Availability Statement and it has passed the technical check (c.f. below) 

5. Is the manuscript presented in an intelligible fashion and written in standard English?

Reviewer #3: No

6. Review Comments to the Author

Reviewer #3: Comments were partially addressed but the document preparation is not following PLoS one criteria. figures and tables were not correctly cited and not well prepared.

Response: The manuscript has passed the basic formatting requirements of PLOS ONE, as confirmed by Oriel Jerome Delas Alas Vida, PLOS ONE on May 17, 2022 (see word document "Response to Reviewers"). To our knowledge, this implies that figures and tables are cited according to PLOS formatting guidelines, that the tables are formatted according to PLOS formatting guidelines, that the figures have passed the Preflight Analysis and Conversion Engine (PACE) digital diagnostic tool, https://pacev2.apexcovantage.com/ and that the data availability statement has been checked. 

it needs revision and grammar errors are serious problems.

Response: We would appreciate quotation of the alleged grammar errors. The manuscript has undergone thorough proofreading by a native speaker who is teaching English at university level (https://www.fhnw.ch/en/people/margaret-oertig). The first author is bilingual and has published in international peer review journals such as American Journal of Epidemiology, Environmental Health Perspectives etc. 

the flow of ideas, neatness, result reporting form, and discussion are not as such good. statistical analysis is not clearly stated and does not answer the objective.

generally, the way that the manuscript is prepared is difficult to understand.

Response: We would have appreciated specific comments on why the reviewer #3 considers the flow of ideas, neatness, and result reporting form “as such not to be good”. We do clearly state statistical analysis, following up-to-date recommendations (lines 303 ff.). We apply appropriate statistical methods to answer the research questions, stated at lines 135-146. We would have appreciated a detailed comment on why the reviewer #3 has difficulties to understand the manuscript and on how we could improve the paper. To clarify the context of the current analysis we have extended the paragraph on the SIHOS study and added two recent publications that made use of the SIHOS-database (lines 125-127): 

“The SIHOS study investigated social disparities that may manifest at different stages of a hospitalization: before hospital admission [3], during the hospital stay (this analysis), at discharge [47] or after the hospital stay [48].”

7. PLOS authors have the option to publish the peer review history of their article (what does this mean?). If published, this will include your full peer review and any attached files.

If you choose “no”, your identity will remain anonymous, but your review may still be made public.

Do you want your identity to be public for this peer review? For information about this choice, including consent withdrawal, please see our Privacy Policy.

Reviewer #3: No

---

## [Editor Report · Decision Letter 2]

18 Jul 2022

Social inequalities, length of hospital stay for chronic conditions and the mediating role of comorbidity and discharge destination: A multilevel analysis of hospital administrative data linked to the population census in Switzerland

PONE-D-21-31664R2

Dear Dr. Bayer-Oglesby,

We’re pleased to inform you that your manuscript has been judged scientifically suitable for publication and will be formally accepted for publication once it meets all outstanding technical requirements.

Kind regards,

Dong Keon Yon, MD, FACAAI

Academic Editor

PLOS ONE

Additional Editor Comments (optional):

This is an excellent and mesmerizing paper!
---

## [Editor Report · Acceptance letter]

3 Aug 2022

PONE-D-21-31664R2 

Social inequalities, length of hospital stay for chronic conditions and the mediating role of comorbidity and discharge destination: A multilevel analysis of hospital administrative data linked to the population census in Switzerland 

Dear Dr. Bayer-Oglesby:

I'm pleased to inform you that your manuscript has been deemed suitable for publication in PLOS ONE. Congratulations! Your manuscript is now with our production department. 

Kind regards, 

on behalf of

Dr. Dong Keon Yon 

Academic Editor

PLOS ONE